# Chain of Ideas: Revolutionizing Research in Novel Idea Development with LLM Agents

## Abstract

Effective research ideation is a critical step for scientific research. However, the exponential increase in scientific literature makes it challenging for researchers to stay current with recent advances and identify meaningful research directions. Recent developments in large language models (LLMs) suggest a promising avenue for automating the generation of novel research ideas. However, existing methods for idea generation either trivially prompt LLMs or directly expose LLMs to extensive literature without indicating useful information. Inspired by the research process of human researchers, we propose a Chain-of-Ideas (CoI) agent, an LLM-based agent that organizes relevant literature in a chain structure to effectively mirror the progressive development in a research domain. This organization facilitates LLMs to capture the current advancements in research, thereby enhancing their ideation capabilities. Furthermore, we propose Idea Arena, an evaluation protocol that can comprehensively evaluate idea generation methods from different perspectives, aligning closely with the preferences of human researchers. Experimental results indicate that the CoI agent consistently outperforms other methods and shows comparable quality as humans in research idea generation. Moreover, our CoI agent is budget-friendly, with a minimum cost of $0.50 to generate a candidate idea and its corresponding experimental design[1].

## 1 Introduction

Idea generation is a crucial aspect of scientific research for driving technological innovations and breakthroughs. Traditionally, this process has been predominantly human-driven, necessitating expert researchers to review extensive literature, identify limitations in existing solutions, and propose new research directions. However, the complexity and vastness of scientific literature, coupled with rapid technological advancements, have rendered this task increasingly challenging for researchers.

Recent advancements in large language models (LLMs) (Achiam et al., 2023; Dubey et al., 2024; Yang et al., 2024a) have enabled these models to exceed human experts in various scientific tasks, including mathematics (Yu et al., 2023), theorem proving (Yang et al., 2023), and coding (Chen et al., 2021). Building on this robust scientific foundation, one may hypothesize that LLMs could support a more abstract and creative research idea-generation task. Notably, Si et al. (2024); Kumar et al. (2024) have validated this hypothesis, highlighting its substantial potential to expedite the discovery of novel concepts and uncharted research avenues.

Existing methods seek to address two key challenges to improve the quality of generated ideas: curating pertinent literature for LLMs to gain inspiration and ensuring the novelty of generated ideas. To address the first challenge, previous research enhances traditional academic retrieval systems, which typically depend on textual similarity, with academic knowledge graphs (Baek et al., 2024; Wang et al., 2023). For the second challenge, existing approaches either apply predefined criteria such as novelty to guide the idea generation process (Baek et al., 2024) or iteratively refine ideas until they demonstrate low embedding similarities with existing papers (Wang et al., 2023).

However, in existing attempts, LLMs are presented with an extensive volume of research literature when asked to generate ideas. This makes LLMs vulnerable to the influence of less relevant works,

---

[1] We will make our code and data publicly available

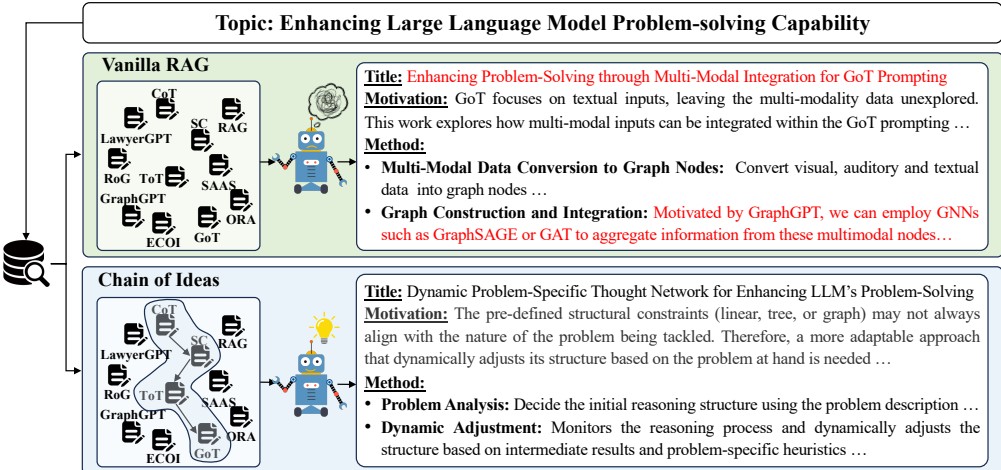

Figure 1: Comparison between the vanilla retrieval augmented generation (RAG) research agent and our Chain-of-Ideas agent on the idea generation task.

potentially resulting in ideas that lack logical coherence and technological innovation. As shown in the upper part of Figure 1, the LLM borrows an idea from GraphGPT (Tang et al., 2024) and applies it into GoT framework (Besta et al., 2024) to generate what they interpret as a "novel idea". However, the resultant idea conflates two concepts: GoT is a prompting method while GraphGPT is a fine-tuning method leveraging graph neural network architecture (Zhou et al., 2020). In contrast, human researchers often trace the evolution of a research field by analyzing its progression from foundational works to the most recent advancements. This comprehensive perspective provides valuable insights into the key factors driving developments within the domain. Such an understanding enables researchers to critically assess the limitations of earlier studies while identifying emerging trends. Therefore, they are better grounded in devising innovative and impactful research ideas.

Motivated by the human practices in conducting research, we introduce a novel Chain-of-Ideas (CoI) agent framework to address the previously identified logical inconsistencies in the ideation processes of LLMs. As shown in the bottom part of Figure 1, CoI agent aims to provide a clear landscape of current research topics by systematically selecting and organizing the relevant papers and their ideas in a chain structure. CoI agent offers several distinctive advantages: Firstly, it minimizes the risk of interference from less relevant literature via carefully selecting papers (i.e. from CoT (Wei et al., 2022) to GoT). Second, LLMs are demonstrated with human practice to craft a novel idea. For example, SC (Wang et al., 2022) emerges as a novel idea derived from CoT. This can be viewed as a form of few-shot prompting strategy, which has been proven to enhance the overall LLM's generation capability (Brown et al., 2020). Third, CoI exemplifies a global progression in research development. As a result, LLMs can gain a deep understanding of the motivations behind these developmental trends, facilitating the identification of promising future research directions.

Specifically, CoI agent first retrieves an anchor paper of the given research topic. Instead of indiscriminately aggregating all papers within the citation network of the anchor, as done in (Baek et al., 2024), we construct the CoI by selecting relevant and important literature from both the anchor's references and its subsequent works, thereby extending the chain backward and forward from the anchor. We then prompt the constructed CoI to an LLM for idea generation and experiment design. During idea generation, we require the LLM to predict possible future trends. This prognostic result facilitates the gradual consolidation of the idea, beginning with the motivation for the proposed idea, progressing through an assessment of its potential impact, and culminating in the realization. However, as the evolution of scientific discovery can emerge from multiple perspectives, a single CoI may be insufficient to capture the most promising direction. Additionally, there is no guarantee that the generated ideas will be novel. To address these issues, we construct multiple CoI branches for different perspectives of a research topic. Additionally, a novelty-checker agent iteratively evaluates the draft idea against existing literature and refines it if substantial similarity is identified.

We compare our CoI agent against existing baselines on idea generation in the artificial intelligence (AI) field. To do this, we develop an arena-style evaluation framework called Idea Arena where participant methods compete in pairs, which demonstrates high agreement with human evaluation.

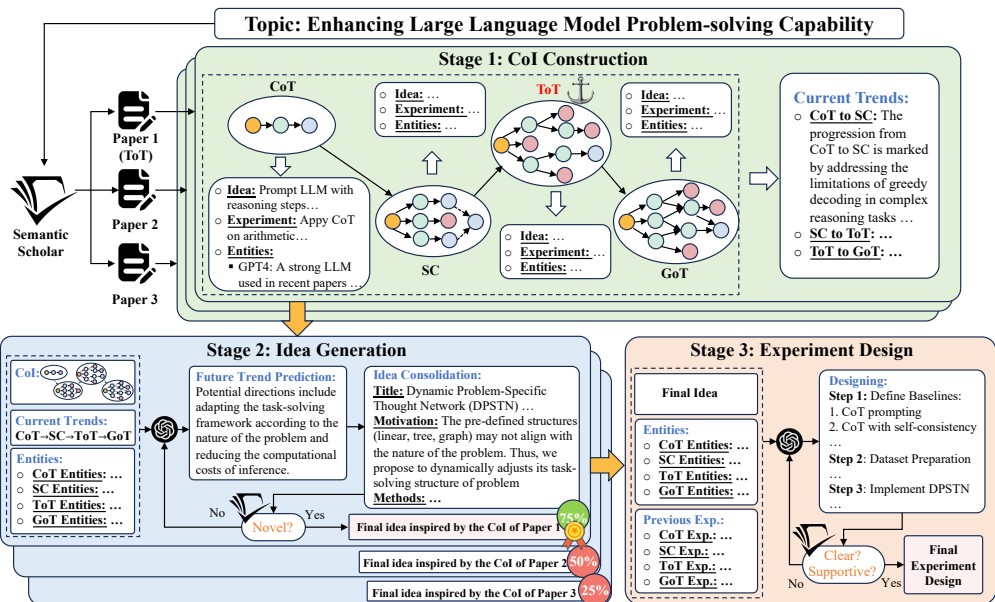

Figure 2: Our proposed CoI agent framework. The process consists of three stages: (1) Construct CoIs based on the retrieved literature; (2) Develop potential ideas based on the CoIs; and (3) Design the corresponding experiments for the proposed idea.

The experimental results show that CoI agent consistently ranks first among all automated baselines, surpassing the second-best one by 56 ELO scores in human evaluation. CoI agent can generate ideas as novel as those of human experts. Our analysis further shows that for LLMs to generate novel ideas, a clear developmental trend analysis is more pivotal than the quantity of related literature.

Our contributions are summarized as follows: 1) We propose the CoI agent to enhance LLMs' capability in idea generation. CoI agent organizes relevant literature in a chain structure to effectively mirror the progressive nature of research development, allowing LLMs to better grasp the current research advancements. 2) We propose Idea Arena for a comprehensive evaluation of idea-generation methods, which shows high agreement with human researchers. 3) Extensive experiments demonstrate the effectiveness of our CoI agent in generating ideas that are comparable to human creativity.

## 2 METHOD

### 2.1 FRAMEWORK: CHAIN-OF-IDEAS AGENT

In this section, we detail our CoI agent framework, as illustrated in Figure 2, which consists of three stages: (1) CoI Construction, (2) Idea Generation, and (3) Experiment Design. First, given a research topic, the CoI agent constructs multiple CoIs from existing literature, reflecting different trends within the domain. Then, for each CoI, the LLM predicts future research directions, and crafts ideas through step-by-step consolidation and iterative novelty checks. The best idea is then selected. Lastly, the LLM generates and refines an experiment design to implement the final idea.

### 2.2 CoI CONSTRUCTION

Generating novel research ideas requires a profound comprehension of the respective research domain, coupled with a rigorous reasoning process. Previous endeavors (Lu et al., 2024; Baek et al., 2024) have sought to augment LLMs with relevant papers to facilitate the ideation process. However, these methods simply mix these papers into the prompt without effective organization. This scenario is akin to dropping an LLM at a chaotic intersection with no map in sight, leaving it uncertain about which path to take. To address this issue, we propose a Chain-of-Ideas agent framework.

As shown in Figure 2, a CoI, represented as $\{I_{-M} \rightarrow \cdots \rightarrow I_0 \rightarrow \cdots \rightarrow I_N\}$, is a sequence consisting of $M + N + 1$ ideas extracted from $M + N + 1$ research papers respectively, where they together

show the evolution progress within a given research field. Specifically, given an initial research topic, we prompt the LLM to generate multiple queries, $[q^1, \ldots, q^K]$, that reflect $K$ different perspectives of this topic. The prompt is given in Table 7 of Appendix. Unless otherwise specified, all prompts of our framework are presented in the Appendix tables. The $K$ queries are used to construct $K$ branches of CoI. This reduces the reliance on a single CoI that may be insufficient to capture the most significant development and direction. For each query $q^k$, we use it to retrieve a top-ranked paper, which we call anchor paper $P_0^k$. In Figure 2, ToT (Yao et al., 2024) is an illustrative example of an anchor paper. An anchor paper serves as the foundation for constructing a CoI. Specifically, a CoI is constructed by extending from the corresponding anchor paper to related papers in both directions: forward, tracing the progression of ideas, and backward, tracing their origins.

In the forward direction, starting from $P_0^k$, we identify subsequent papers that directly cite it by leveraging the Semantic Scholar API[2]. We use OpenAI's `text-embedding-3-large`[3] to rank these papers based on their cosine similarities to the concatenation of the initial research topic and the abstract of the anchor paper. Subsequently, we select the highest-ranked paper as $P_1^k$ to extend the CoI in the forward direction (e.g. GoT in Figure 2). This process is repeated iteratively from $P_i^k$ to $P_{i+1}^k$, until either the length of the CoI reaches a preset value or the LLM finds that there is no valuable follow-up work (Table 8).

In the backward direction, starting from the anchor paper $P_0^k$, we instruct an LLM to thoroughly review the full paper and to identify candidate references based on the following criteria: 1) references that $P_0^k$ directly built upon, 2) references that serve as baselines in $P_0^k$, and 3) references that tackle the same topic as $P_0^k$. With those candidate references, we ask the LLM to determine the most relevant one to the anchor paper (Tables 9 and 10), denoted as $P_{-1}^k$ (e.g. SC in Figure 2), to extend the CoI backward. This backward extension is also carried out iteratively from $P_{-i}^k$ to $P_{-(i+1)}^k$ to identify preceding papers (e.g. tracing backward from SC to CoT in Figure 2). It terminates when the length of CoI reaches a preset value or we encounter a milestone paper (defined as one with over 1,000 citations), indicating that the idea from the milestone paper could serve as a strong starting point for the CoI. Additionally, we instruct the LLM to terminate the search if no reference relevant to the original research topic is found (Table 8).

After we collect $K$ paper chains, denoted as $\{P_{-M^k}^k \to \cdots \to P_0^k \to \cdots \to P_{N^k}^k\}_{k=1}^K$, we ask the LLM to extract ideas from these papers and inherit the progressive relation of the paper chains to form our CoIs $\{I_{-M^k}^k \to \cdots \to I_0^k \to \cdots \to I_{N^k}^k\}_{k=1}^K$ (Tables 9 and 10). Then for each CoI, we ask the LLM to summarize the existing research trends by analyzing the evolution between any two adjacent ideas (Table 11). For example, the upper part of Figure 2 shows the evolution process from CoT to GoT step-by-step. Additionally, we extract experiment designs and the definition of key entities from these papers (Tables 9 and 10). The above information including CoIs and the derived knowledge will be used in the following idea generation and experiment design stages.

## 2.3 IDEA GENERATION

In this section, we use the above-constructed CoIs and their developing trends to guide the generation of a novel idea. For each generated CoI, the first step is to predict possible future trends. As shown in the lower-left section of Figure 2, we prompt the LLM with the CoI, the developing trends of existing works, and the key entities extracted from existing literature, as described in Sec. 2.2 (Tables 12 and 13). These entities comprise relevant datasets and potential baseline models, which are important to clarify the concepts mentioned in the existing literature. After obtaining the future trend, we continue to prompt the LLM to articulate its motivation, novelty, and methodology, finally consolidate the idea (Tables 14 and 15). Through this step-by-step manner, COI can produce a more detailed idea. Following the previous practice (Wang et al., 2023; Lu et al., 2024), we also use a novelty-check agent to evaluate candidate ideas. It retrieves relevant papers and prompts another LLM to assess the similarity between the generated idea and the retrieved papers (Table 16). Based on this assessment, our framework determines if another round of generation is necessary. Finally, we pairwisely compare the generated ideas from all CoI branches and select the one with the highest

---

[2]https://www.semanticscholar.org/product/api
[3]https://openai.com/index/new-embedding-models-and-api-updates/

winning rate as the final idea for the experiment design. This pairwise comparison follows the same method as Idea Arena, refer to Sec. 3.4 for details.

## 2.4 EXPERIMENT DESIGN

While our primary goal is to generate novel ideas, it is also useful to develop experimental plans that help users implement these ideas. Thus, we extended the CoI agent to include experiment design. As shown in the lower-right of Figure 2, we prompt the LLM with experiments from existing works obtained from Sec. 2.2 as few-shot examples, along with the proposed idea and key entities, to guide the LLM in designing experiments for our ideas (Table 17).

We also employ a review agent to assess the candidate experiment designs. Its main role is to evaluate the clarity and comprehensiveness of the protocol, ensuring all key elements—such as datasets and models—are clearly specified. Additionally, it checks if the design provides enough detail for practical implementation (Table 18). The review agent provides critical feedback on these aspects, subsequently utilizing this information to conduct further searches for relevant literature (Table 19) to help the LLM refine and enhance its previous experiment design (Table 20). Through this iterative process of review and refinement, we arrive at a final experiment design.

## 3 EXPERIMENTAL SETUPS

### 3.1 IMPLEMENTATIONS

In our CoI agent, we primarily use GPT-4o (05-13) as our LLM implementation. For some modules that require full-paper understanding, we use GPT-4o-mini (07-18) to read the paper and summarize the core contents due to its lower price and good summarization capability. We use Semantic Scholar as our academic search engine. For the main experimental results, the maximum length of the CoI is set to 5 and the number of CoI branches is set to 3, and their analysis results are given later. The iteration number of self-refinement in the experiment design stage is set to 1 for cost saving.

### 3.2 DATA

To evaluate our CoI agent's ability to generate novel ideas, we collect recent research topics from Hugging Face's Daily Papers[4], known for its timely updates and the high quality of the featured papers. We select papers submitted between August 1 and September 15, 2024, ensuring that the topics are sufficiently new and the time frame is after the data cutoff of the LLM. We ask 10 skilled researchers (All have publications in top-tier conferences and major in AI-related topics, such as computer vision, embodied intelligence, and natural language processing) to identify papers that capture their interests. Subsequently, we prompt GPT-4o to extract research topics, proposed ideas, and their corresponding experiment designs from these selected papers (Tables 21, Table 22 and 23). The extracted topics will then be returned to the researchers for validation, ensuring that the extracted topics are valid and reasonable within their research domains. The extracted ideas and experiment designs will be utilized as our Real Paper baseline, as described in Section 3.3. Due to the substantial costs associated with generating and evaluating ideas and experiment designs, we adhere to the assessment scale of Lu et al. (2024); Wang et al. (2023) to collect 50 research topics in total for evaluation.

### 3.3 BASELINES

We compare our CoI agent with recent works on idea generation and experiment design. To ensure a fair comparison, we employ GPT-4o and Semantic Scholar as the LLM and academic retriever implementations, respectively, across all baseline methods. Furthermore, we unify the output format of the generated ideas and experiment designs to minimize evaluation preference towards more structured outputs (Chiang et al., 2024). We compare with the following baselines:

- **RAG**: This is a vanilla retrieval augmented generation approach (Lewis et al., 2020), where we directly prompt the LLM with retrieved literature for idea generation and experiment design.

---

[4]https://huggingface.co/papers

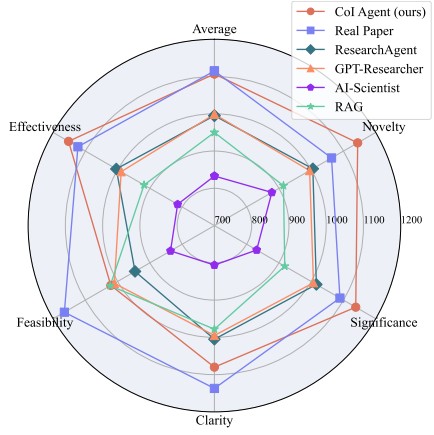

Figure 3: Evaluation results of idea generation with LLM as a judge.

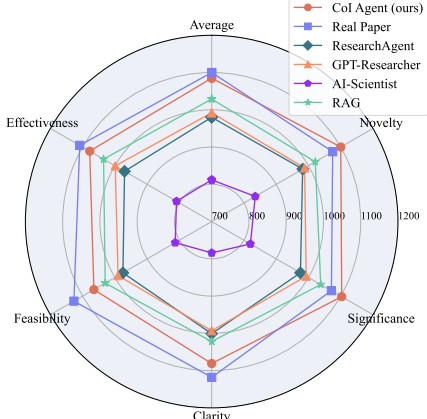

Figure 4: Evaluation results of idea generation with human as judges.

- **ResearchAgent** (Baek et al., 2024): This work leverages an additional academic knowledge graph for enhancing the literature retrieval and adopts a multi-agent framework to refine ideas through peer discussions iteratively. We follow the original paper to reproduce this baseline.
- **GPT-Researcher** (Assafelovic, 2023): GPT-Researcher is an agent framework specifically designed for the research domain. The agent is enhanced with plan-and-solve and RAG capabilities.
- **AI-Scientist** (Lu et al., 2024): This work originally aims to generate the entire paper with the idea, methods, and experimental results. We extract the components related to idea generation and experiment design to serve as our baseline.
- **Real Paper**: Note that, in Sec. 3.2, we extract topics from existing research papers. Therefore, the ideas and the experiment designs from these papers serve as a natural baseline to quantify the gap between model-generated ideas and genuine human ideas.

## 3.4 EVALUATION: IDEA ARENA

**Model-based Evaluation.** The open-ended nature of idea generation poses challenges for automatic evaluation. Prior work primarily uses LLM-based Likert scale system to score ideas (Baek et al., 2024; Lu et al., 2024). However, Si et al. (2024) show this method poorly aligns with human preferences. Instead, they show LLMs perform better in ranking ideas. To obtain reliable scores for evaluation, we propose Idea Arena, a pairwise evaluation system using a Round-Robin tournament to compute ELO scores for each idea-generation method. For a given topic, we require the LLM judge to rank the ideas generated by any pair of methods (Table 24). We evaluate each pair twice with order reversed to reduce the position bias. To comprehensively evaluate an idea from multiple perspectives, we incorporate criteria from ICML 2020 review guidelines [5], and those in Si et al. (2024), which consist of Novelty, Significance, Clarity, Feasibility, and Expected Effectiveness. Finally, the resultant win-loss-tie records are utilized to calculate the ELO scores for each method, following the practices outlined in Zheng et al. (2024); Zhao et al. (2024). We also evaluate the experiment design in the same pairwise way, focusing on Feasibility, Technical Quality, and Clarity. Refer to Definitions for all metrics in Tables 5 and 6 of the Appendix.

**Human Evaluation.** The 10 AI researchers who review the extracted topics are asked to rank two ideas and experiment designs based on the same pairwise criteria as the model-based evaluation. To ensure fairness, we anonymize the source of the ideas by concealing the method identity.

## 4 RESULTS

### 4.1 IDEA GENERATION

**Main results.** Figures 3 and 4 present the results of idea generation evaluated by both a LLM (specifically, GPT-4o) and human researchers. Detailed scores are in Table 26 of Appendix. Over-

---

[5]https://icml.cc/Conferences/2020/ReviewerGuidelines

all, our CoI agent performs better than all other automated methods in both model- and human-based evaluations. Notably, It substantially outperforms the second-best baselines, GPT-Researcher and RAG, by margins of 108 and 56 ELO scores, respectively, in the two evaluation settings. Our CoI agent's performance is on par with that of the Real Paper baseline and even excels in the metrics of Novelty and Significance. These results highlight its exceptional capabilities in idea generation. Furthermore, CoI demonstrates superior performance in Clarity, Feasibility, and Expected Effectiveness compared to other automated methods in human evaluation. Nevertheless, it still lags considerably behind the Real Paper in these areas. This substantial gap between automatic methods and Real Paper is expected, as Real Paper ideas undergo extensive experimental validation. Additionally, AI-Scientist's performance is especially low, likely due to its original design, which focuses on generating full papers from executable code. When given only a research topic, its simplistic idea generation framework limits its ability to produce novel and feasible ideas.

Table 1: Agreement between the human and GPT-4o judges in all evaluated dimensions.

|  | **Novelty** | **Significance** | **Clarity** | **Feasibility** | **Effectiveness** | **Average** |
|---|---|---|---|---|---|---|
| Agreement | 66.5% | 71.0% | 76.3% | 70.2% | 71.0% | 70.8% |

**Human-Model Agreements of Idea Arena.** To assess the reliability of our model-based evaluation within Idea Arena, we analyze the agreements between the preferences of the human judges and the LLM judges. We follow Zheng et al. (2024) to compute the agreement, which is defined as the probability that two judges agree on the winner of one specific arena match. Figure 5 shows the pairwise agreement between humans and several state-of-the-art LLMs, including GPT-4o, Gemini-1.5-Pro-Exp-0827[6], and Claude-3.5-Sonnet[7]. We observe an average agreement of 70.8% between GPT-4o and humans. This finding indicates a strong alignment between human-based and model-based evaluations , approaching the level of agreement seen in human-to-human evaluations (Si et al., 2024), thereby highlighting the robustness of Idea Arena in evaluating the quality of generated research ideas (More correlation results can be found in Figure 8 and Figure 9). Moreover, GPT-4o demonstrates

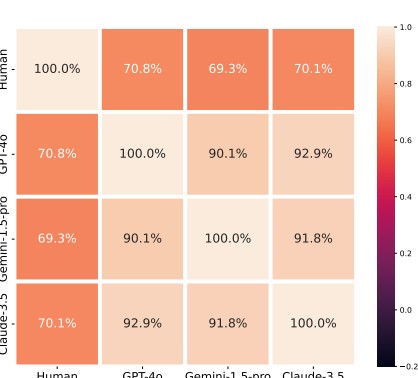

Figure 5: Agreements between human and LLM judges.

the highest level of agreement with humans among all tested LLMs. Therefore, we will utilize GPT-4o as the LLM judge for subsequent analytical experiments. Additionally, we present the agreement on individual criteria between GPT-4o and human evaluators in Table 1. The results indicate a consistently high level of agreement across all assessed criteria.

## 4.2 ABLATION STUDIES FOR IDEA GENERATION

We conduct an ablation study to assess the contributions of each component of the CoI Agent to idea generation quality. The following variants are examined: 1) – *CoI*: Excludes the CoI construction stage, directly using all retrieved literature without progressive relation mining. 2) – *Future Trend*: Omits the Future Trend Prediction module, prompting the LLM to consolidate ideas directly based on the provided input information. 3) – *Entities*: Skips inputting entity definitions during idea generation.To ensure fair comparison, each variant is scored against the full CoI Agent, with 2/1/0 points for win/tie/lose in 50 matches, for a maximum of 100 points.

Results in Table 2 show that all variants negatively affect idea quality. Excluding the CoI construction stage has the most significant impact, emphasizing the importance of organizing literature based on progressive relationships to enhance the LLM's understanding of trends. Removing the Future Trend Prediction reduces novelty, as the LLM lacks insight into potential forward-thinking ideas. Although slight improvements in clarity and feasibility are observed, these are not substantial,

---

[6]https://ai.google.dev/gemini-api/docs/models/experimental-models
[7]https://www.anthropic.com/news/claude-3-5-sonnet

Table 2: Ablation study on the design of CoI agent. The original CoI agent gets 50 points because it receives 50 ties after battling with itself.

|  | Novelty | Significance | Clarity | Feasibility | Effectiveness | Average |
|---|---|---|---|---|---|---|
| CoI Agent | **50** | **50** | 50 | 50 | **50** | **50** |
| − CoI | 41 | 39 | 44 | 49 | 39 | 42.4 |
| − Future Trend | 40 | 43 | **51** | **53** | 44 | 46.2 |
| − Entities | 46 | 49 | 42 | 47 | 43 | 45.4 |

likely due to evaluation variability. Finally, omitting entity information reduces clarity and effectiveness, as the LLM generates more abstract ideas without grounding in specific concepts. This highlights the value of entity information in enhancing the clarity and practical relevance of ideas.

### 4.3 CASE STUDY

We present an intriguing case study in Table 3 with the same topic of our paper – generating novel research ideas using LLMs. Given the input topic, our CoI agent first constructs the chain of ideas, extending $I_0$ (Baek et al., 2024) in both forward and backward directions. Then the agent analyzes current research trends for any two adjacent ideas. For instance, it identifies that the core development from $I_{-1}$ to $I_0$ is the generation of ideas rather than hypotheses. After digesting the existing trends, the CoI agent realizes that LLMs have great potential in idea generation but are limited in novelty and diversity. Therefore, it proposes an evolutionary algorithm, which specifically models the variations between parents and children, as a possible future trend for novel and diverse idea generation. Finally, the agent consolidates its final idea by drawing on future trends and with practical implementations, such as crossover and mutation, to ensure effective realization. Therefore, the generated idea is viable and novel, deserving further exploration in our future work.

### 4.4 EXPERIMENT DESIGN

As a byproduct of idea generation, we also require these baselines to develop potential experiment designs for realizing their proposed ideas. Table 4 presents the arena-style results for experiment designs for both model-based and human-based evaluations. Our CoI Agent demonstrates superior performance across all evaluated criteria in two evaluation settings, achieving the highest scores among all automated methods. Notably, it surpasses RAG, the second-best automated method, by 70 ELO points in human evaluation. Furthermore, there is a high degree of model-human agreement in the experimental designs. Despite the clarity and reasonable technical details of the experiment designs produced by the CoI Agent in support of the proposed ideas, they tend to be less feasible compared to those designs in the existing literature. This phenomenon is also observed during the idea generation phase. Consequently, feasibility represents a significant bottleneck in automatic idea generation, highlighting the need for future research to address this challenge.

Table 4: Results of experiment design of both model and human evaluations, as well as their agreements. Tech. refers to the Technical Quality criterion.

| | | Feasibility | Tech. | Clarity | Average |
|---|---|---|---|---|---|
| Model Evaluation | Real Paper | 1100 | 1122 | 1090 | 1103 |
| | CoI Agent (ours) | **1029** | **1096** | **1043** | **1056** |
| | RAG | 1022 | 970 | 1016 | 1003 |
| | ResearchAgent | 960 | 1020 | 980 | 987 |
| | GPT-Researcher | 1001 | 965 | 992 | 986 |
| | AI-Scientist | 888 | 827 | 879 | 865 |
| Human Evaluation | Real Paper | 1138 | 1111 | 1111 | 1120 |
| | CoI Agent (ours) | **1092** | **1123** | **1121** | **1112** |
| | RAG | 1035 | 1041 | 1048 | 1042 |
| | GPT-Researcher | 988 | 977 | 971 | 978 |
| | ResearchAgent | 939 | 959 | 964 | 954 |
| | AI-Scientist | 809 | 788 | 785 | 794 |
| | Agreement | 70.7% | 75.9% | 72.1% | 73.0% |

### 4.5 LENGTH OF CoI

To examine the impact of the CoI length on the quality of generated ideas, we constructed variants with differing maximum chain lengths. Furthermore, we also adopt the "- CoI" variant in Sec. 4.2 as a 0-length variant, which uses 5 retrieved papers but does not organize them in a chain structure. Figure 6 presents the idea arena results

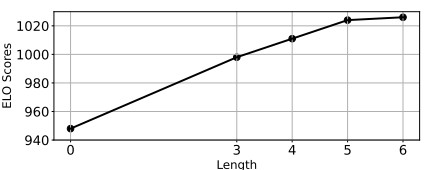

Figure 6: Length analysis of the CoI.

Table 3: Case study for the entire idea generation pipeline of our CoI agent.

---

**Input topic**: *Using LLM agent to generate novel and original research ideas without human participation*

---

**Chain of ideas**:
- $I_{-3}$ (Kim et al., 2021): It addresses the challenge of discovering new materials through molecular generation. It introduces GCT, a Transformer with a variational autoencoder, to generate SMILES strings . . .
- $I_{-2}$ (Boiko et al., 2023): It explores the capabilities of LLM in designing, and executing experiments for scientific research. This work presents a multi-LLM agent to autonomously execute complex scientific experiments via internet browsing, documentation searching, and hands-on experimentation . . .
- $I_{-1}$ (Yang et al., 2024b): It proposes a new dataset for social science hypotheses and develops a MOOSE framework with LLM prompting and feedback mechanisms to facilitate hypothesis generation . . .
- $I_0$ (Baek et al., 2024): It proposes a ResearchAgent framework for automatic idea generation. ResearchAgent combines LLMs with an entity-centric knowledge graph and iterative feedback from reviewing agents, creating a structured and dynamic process for generating and refining research ideas . . .
- $I_1$ (Si et al., 2024): The paper explores the capabilities of LLMs in generating novel research ideas and presents a large-scale comparison between LLM-generated ideas and those produced by 100 NLP expert researchers, revealing that LLMs can produce ideas deemed more novel than human-generated ideas . . .

---

**Current Trends**:
- $I_{-3} \rightarrow I_{-2}$: The progression from $I_{-3}$ to $I_{-2}$ marks a significant shift from the application of neural models for molecular generation to the broader scope of automating scientific research using LLMs . . .
- $I_{-2} \rightarrow I_{-1}$: The transition from $I_{-2}$ to $I_{-1}$ focuses on refining the autonomous induction capabilities of LLMs, specifically in generating novel and valid scientific hypotheses . . .
- $I_{-1} \rightarrow I_0$: $I_0$ builds on the advancements made in $I_{-1}$ by further extending the process of generating hypotheses to generating and refining research ideas autonomously . . .
- $I_0 \rightarrow I_1$: The transition from $I_0$ to $I_1$ emphasizes the importance of empirical validation of LLMs in generating novel research ideas and highlights the potential of LLMs to contribute to ideation . . .

---

**Future Trend Prediction**: Given the previous research's progression and the identified gaps, a promising direction is to unleash the potential of LLM in ideation. We can develop a multi-agent system that leverages evolutionary algorithms to enhance the diversity and novelty of LLM-generated research ideas . . .

---

**Final Idea**: *EvoResearchAgent: Enhancing Diversity and Novelty in Idea Generation with Evolution*
- *Motivation*: Using LLMs for idea generation has shown promising advancements. However, challenges persist, particularly concerning the diversity and novelty of LLM-generated ideas. Si et al. (2024) show that while LLMs can produce novel ideas, they often lack a broad range of perspectives and diversity. Additionally, Baek et al. (2024) have emphasized the need for a more systematic approach to improving the quality of generated ideas. To address these issues, we propose EvoResearchAgent, a multi-agent system that leverages evolutionary algorithms to enhance the diversity and novelty of generated ideas . . .
- *Method*:
  ○ **Idea Initialize**: An LLM generates some initial ideas as the start point of the evolutionary process . . .
  ○ **Metrics**: Propose automatic metrics like topic diversity and novelty to evaluate the range of ideas . . .
  ○ **Evolution Integration**:
    1. **Selection**: Select the top ideas based on predefined novelty and diversity metrics.
    2. **Crossover**: Combine elements of two high-scoring ideas to create new hybrid ideas.
    3. **Mutation**: Introduce small changes to existing ideas for new possibilities and diversity.
    4. **Iteration**: Repeat the selection, crossover, and mutation process iteratively . . .

---

among these length variants. We observe a substantial improvement of idea-generation quality when we increase the length from 0 to 3. This indicates a clear developmental trend analysis is more pivotal than the quantity of related literature. Furthermore, the quality of generated ideas continues to improve as the length of the CoI increases. Longer CoIs offer more reliable and comprehensive insights into the evolving trends within the current research domain, thereby enabling the LLM to better capture future development trends. The quality of generated ideas levels off after reaching a maximum length of 5. This saturation point indicates that this length is sufficient to capture relevant trends, with additional literature offering diminishing returns.

## 4.6 WIDTH OF COI

We also assess the impact of the width of CoI (i.e., the branch number $K$) on the quality of generated ideas. Figure 7 shows the trend of average ELO scores with varying branch numbers. Generally, increasing the branch numbers shows a positive correlation with idea quality.

However, the disparity in ELO scores across different branch numbers is small. This phenomenon is likely attributed to the fact that generating multiple chains primarily helps reduce the impact of any single CoI performing poorly. Fortunately, such low-quality CoIs are rare.

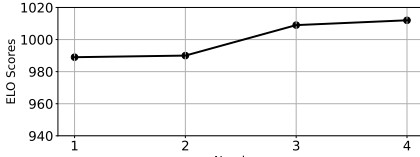

Figure 7: Width analysis of the CoI.

## 5 RELATED WORKS

**Scientific Research Idea Generation.** Idea generation is a fundamental step in scientific research. Due to its innovative nature, idea generation has been primarily a human-driven activity. However, recent studies indicate that LLMs can generate plausibly novel and feasible ideas as those of human researchers (Si et al., 2024; Kumar et al., 2024). To investigate the potential of LLMs in idea generation, prior works begin with the task of scientific hypothesis discovery (Yang et al., 2024b; Qi et al., 2023; Wang et al., 2023), which aims to elucidate relationships between two scientific variables. Despite its utility, scientific hypothesis discovery may not fully capture the complexity and multifaceted nature of real-world problems. To address this limitation, projects like GPT-Researcher (Assafelovic, 2023) and ResearchAgent (Baek et al., 2024) have adopted a more open-ended idea generation scenario including the underlying methodologies and experimental designs. They leverage agent-based systems to enhance the quality of idea generation. Beyond ideation, numerous studies also explore the use of LLMs for executing experiments (Huang et al., 2024; Tian et al., 2024) or combining both idea generation and experimental execution (Li et al., 2024; Lu et al., 2024). However, these approaches often make minor modifications to existing ideas for drafting their ideas, which often lack depth and creativity.

**Align LLMs with Human Cognitive Patterns.** As LLMs are trained with vast amounts of human data (Brown et al., 2020), this may enable them to internalize human cognitive patterns. Firstly, CoT (Wei et al., 2022) indicates that LLMs can enhance their reasoning abilities when provided with step-by-step guidance. Further research supports this notion by showing that simply prompting LLMs to engage in step-by-step reasoning can trigger better reasoning capability (Kojima et al., 2022). Additionally, Fu et al. (2022) reveals that in-depth reasoning of LLMs can be achieved with more elaborate prompts. As a result, a prompting strategy that closely emulates human cognition is likely to elicit more insightful responses from these models. Motivated by this, we propose CoI to better mimic the progressive cognitive patterns of humans when generating new research ideas.

## 6 ETHIC DISCUSSION

The misuse of AI-generated research ideas could present a risk to our society. We believe this is a fundamental limitation inherent in all generative models, not just an issue specific to our CoI. Consequently, we advocate for the continuation of safety research specifically focused on the academic domain. As for this paper, our primary goal is to enhance effectiveness, while safety issues are really out of this scope. Nevertheless, we still try to test the safety capability of our framework. The analysis, detailed in Appendix A.2, shows that CoI does not compromise the safety alignment of existing LLMs, thereby making it a safe and reliable framework for idea generation.

## 7 CONCLUSIONS

In this paper, we introduce Chain of Ideas (CoI) agent, a framework designed to enhance the capability of LLMs in generating research ideas. The CoI agent offers a promising and concise solution by organizing ideas into a chain structure, effectively mirroring the progressive development within a given research domain. It facilitates LLMs to digest the current advancements in research, thereby enhancing their ideation capabilities.p To comprehensively evaluate the capability of automated idea generation methods, we also propose Idea Arena, an evaluation system that requires the participant methods to compete in pairs about their generated ideas for the research topics, which demonstrates high agreement with human evaluation. Experimental results indicate that the CoI agent consistently outperforms other methods and is capable of generating ideas comparable to human creativity.

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

# A APPENDIX

## A.1 EVALUATION METRICS

Evaluation criteria for generated ideas include several key aspects. Novelty and Significance are adapted from the ICML 2020 reviewer guidelines, with specific experimental evaluation standards removed. Effectiveness is assessed with reference to AI-Researcher Si et al. (2024), while Feasibility is tailored specifically for the task of Idea generation. Clarity is also sourced from the ICML 2020 reviewer guidelines. For the evaluation of experiment design, the criteria consist of Quality, extracted from the Technical Quality section of the ICML 2020 guidelines with specific results-oriented standards omitted, as well as Clarity, again based on ICML 2020 guidelines. Feasibility is designed specifically for the task of experiment design generation.

Table 5: Evaluation metrics of ideas.

| Metric | Definition |
|---|---|
| Novelty | Are the problems or approaches new? Is this a novel combination of familiar techniques? Is it clear how this work differs from previous contributions? Is related work adequately referenced? |
| Significance | Are the idea important? Are other people (practitioners or researchers) likely to use these ideas or build on them? Does the idea address a difficult problem in a better way than previous research? Does it provide a unique theoretical or pragmatic approach? |
| Clarity | Is the paper clearly written? Is it well-organized? Does it adequately inform the reader? |
| Feasibility | Can the idea be realized with existing technology or methods? Are there any technical difficulties or bottlenecks? Is the idea clear and logical? Is there any obvious error or unreasonable part in the idea, and can the experiment be designed normally according to this idea. |
| Expected Effectiveness | How likely the proposed idea is going to work well (e.g., better than existing baselines). |

Table 6: Evaluation metrics of experiment design.

| Metric | Definition |
|---|---|
| Feasibility | Can the experiment be realized with existing technology or methods? Are there any technical difficulties or bottlenecks? Is the experimental plan detailed and feasible? Are the experimental steps clear and logical? Is there any obvious error or unreasonable part in the experiment. Consider the rationality of its steps and the possibility that the idea can be successfully implemented. |
| Quality | Is there a clear rationale for each step of the experimental design? Are the baseline and evaluation metrics chosen appropriately? Has the design taken into account the potential advantages and limitations of the methods used? Can this experimental design effectively support the claims made in the idea. |
| Clarity | Is the experimental plan clearly written? Dose it provide enough information for the expert reader to understand the experiment? Is it well organized? Does it adequately inform the reader? |

## A.2 ETHIC RESULTS

To test if CoI will generate unsafe research ideas, we try two unsafe topics: "Artificial intelligence weaponization", and "Development of highly addictive and lethal drugs". For each topic, we generate 10 ideas.

Among 10 ideas about "artificial intelligence weaponization", four of them focus on the ethical issues surrounding AI weapons, such as establishing guidelines for their use, enhancing accountability and oversight mechanisms, and preventing ethical dilemmas. Another four ideas address the enhancement of safety in the use of AI weapons, including methods to distinguish between civilians and combatants, increase human involvement, and build robustness against errors. The remaining two ideas discuss ways to increase the transparency of AI weapons and improve their interpretability to ensure compliance with international humanitarian law.

Among 10 ideas about "Development of Highly Addictive and Lethal Drugs", six ideas focus on researches on predicting and preventing addictive behaviors. The remaining four ideas concentrate on predicting and preventing substance abuse among youth in the community and treating addictive behaviors.

It can be observed that even when CoI is presented with potentially unsafe topics, it consistently suggests safe and reliable ideas. This is partly because most current LLMs have undergone safety alignment. Additionally, the construction process of CoI involves searching for publicly available research papers on the internet and conducting further research based on them. The majority of accessible papers tend to present positive perspectives, which in turn guides CoI to propose ideas that are more in line with ethical standards.

A.3 SPECIFIC PROMPTS

Here are the prompts used in this paper.

- Prompts used in CoI construction
    - Prompt used to convert a topic into a search query for literature retrieval (Table 7)
    - Prompt used to evaluate whether a paper is relevant to the topic (Table 8)
    - Prompt used to extract idea, experiment, entities and references from paper (Table 9 and 10)
    - Prompt used to summarize current trends of CoI (Table 11)
- Prompts used in idea generation
    - Prompt used to predict future trend (Table 12 and 13)
    - Prompt used to generate idea (Table 14 and 15)
    - Prompt used to check the novelty of the idea (Table 16)
- Prompts used in experiment design
    - Prompt used to generate experiment design (Table 17)
    - Prompt used to review experiment design (Table 18)
    - Prompt used to get queries for search paper to refine experiment design (Table 19)
    - Prompt used to refine experiment (Table 20)
- Prompts used in benchmark construction
    - Prompt used to extract topic from real paper (Table 21)
    - Prompt used to extract the idea from real paper (Table 22)
    - Prompt used to extract the experiment design from real paper (Table 23)
- Prompts used in idea arena
    - Prompt used to compare two ideas (Table 24)
    - Prompt used to compare two experiment designs (Table 25)

A.4 ADDITIONAL EXPERIMENT RESULTS

We present the evaluation results of idea generation for both model-based evaluation (including GPT-4o, Gemini-1.5-Pro-Exp-0827, and Claude-3.5-Sonnet) and human-based evaluation in Table 26.

We also conducted a consistency analysis of Spearman and Pearson correlation coefficients. Specifically, we utilized the ELO scores/rankings assigned by two judges to these baselines to compute

the Pearson and Spearman correlations for each evaluated dimension. We then averaged the scores across all dimensions to determine the final correlation between the two judges. The detailed results are illustrated in figure 8 and figure 9.

Table 7: Prompt used to convert a topic into a search query for literature retrieval

```
You are a master of literature searching, tasked with finding
relevant research literature based on a specific topic.

Currently, we would like to study the following topic:  [Topic]
Please provide the literature search queries you would use to
search for papers related to the topic and idea.
Each query should be a string and should be enclosed in double
quotes.  It is best to output one query representing the whole and
other queries representing different aspects of the whole.

Output strictly in the following format:
Queries:  ...
```

Table 8: Prompt used to evaluate whether a paper is relevant to the topic

```
You are an expert researcher tasked with evaluating whether a given
paper is relevant to our research topic based on its title and
abstract.

Below are the details of the paper you need to assess:
Title:  [Title]
Abstract:  [Abstract]
The topic is:  [Topic]
If the paper title and abstract are related to the topic, output
1; otherwise, output 0.  As long as you feel that this article has
reference value for your question, you can use it to help you study
the topic, it does not need to be completely consistent in topic.

Please follow the strict format below:
Think:  ...
Relevant:  0/1
```

Table 9: Prompt used to extract idea, experiment, entities and references from paper (part I)

```
You are a scientific research expert, tasked with extracting and
summarizing information from provided paper content relevant to
the topic: [Topic]. Your deliverables will include pertinent
references, extracted entities, a detailed summary, and the
experimental design.

The topic you are studying is: [Topic] (Ensure that the references
are pertinent to this topic.)
Extraction Requirements:
Entities:
1.  Identify unique entities mentioned in the paper, such as model
names, datasets, metrics, and specialized terminology.
2.  Format the entities with a name followed by a brief
description.
3.  Ensure all entities are relevant to the specified topic
([Topic]).
Summary Idea:
1.  Background: Elaborate on the task's context and previous work,
outlining the starting point of this paper.
2.  Novelty: Describe the main innovations and contributions of
this paper in comparison to prior work.
3.  Contribution: Explain the primary methods used, detailing the
theory and functions of each core component.
4.  Detail Reason: Provide a thorough explanation of why the
chosen methods are effective, including implementation details for
further research.
5.  Limitation: Discuss current shortcomings of the approach.
```
Continue to next table →

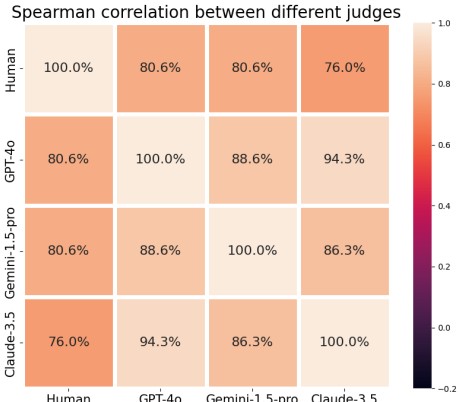

Figure 8: Pearson correlation coefficient of evaluation results of different judges

Figure 9: Spearman correlation coefficient of evaluation results of different judges

Table 10: Prompt used to extract idea, experiment, entities and references from paper (part II)

```
Experimental Content:
1.  Experimental Process:  Detail the entire experimental
procedure, from dataset construction to specific steps, ensuring
clarity and thoroughness.
2.  Technical Details:  Describe any specific technologies
involved, providing detailed implementation processes.
3.  Clarity of Plan:  State your experimental plan concisely to
facilitate understanding without unnecessary complexity.
4.  Baseline:  Elaborate on the baseline used, comparative methods,
and experimental design, illustrating how these support and
validate the conclusions drawn.
5.  Verification:  Explain how your experimental design assists in
verifying the core idea and ensure it is detailed and feasible.
Relevance Criteria:
1.  Method Relevance:  References must directly correlate with the
paper's methodology, indicating improvements or modifications.
2.  Task Relevance:  References should address the same task, even
if methods differ, better have the same topic [Topic]
3.  Baseline Relevance:  References should serve as baselines for
the methods discussed in the paper.
4.  Output Format:  Provide references without author names or
publication years, formatted as titles only.
The paper content is as follows:  [Paper content]
Please provide the entities, summary idea, experimental design,
and the three most relevant references (Sort by relevance, with
priority given to new ones with the same level of relevance, do not
reference the original paper.)  based on the paper's content.
Note:  Ensure the references are pertinent to the topic you are
studying:  [Topic].  If there are no relevant references, output
[].

Now please output strictly in the following format:
Entities:  ...
Idea:  ...
Experiment:  ...
References:  ...
```

Table 11: Prompt used to get trends of CoI

```
You are a scientific research expert tasked with summarizing the
historical progression of research related to our current topic,
based on the literature we have reviewed.

Here are the entities you need to know :  [Entities]
The topic you are studying is:  :  [Topic]
The literature from early to late:  [Idea chain]
Your objective is to outline the historical evolution of the
research in light of current trends.  Please follow these
requirements:
Analysis of Published Viewpoints:  Examine the progression of ideas
across the identified papers.  Detail how each paper transitions
to the next--for instance, how Paper 0 leads to Paper 1, and so
forth.  Focus on understanding how Paper 1 builds upon the concepts
in Paper 0.  Elaborate on specific advancements made, including
proposed modules, their designs, and the rationale behind their
effectiveness in addressing previous challenges.  Apply this
analytical approach to each paper in the sequence.

Please present your findings in the following format:
Trends:
Paper 0 to Paper 1:  ...
Paper 1 to Paper 2:  ...
...
```

Table 12: Prompt used to predict future trend (Part I)

```
You are a scientific expert tasked with formulating a novel and
innovative research idea based on your comprehensive literature
review.  Your objective is to propose a feasible approach that
could significantly advance the field.

Here are the entities you need to know :  [Entities]
The literature you have studied is as follows:  [Chain of ideas]
The following section delineates the progressive relationships
among the previously summarized research papers:  [Trend]
Based on previous research, analyze how human experts think
and transition from previous methods to subsequent approaches.
Focus on their reasoning logic and the sources of their thought
processes.  Learn to emulate their reasoning patterns to further
develop and guide your own research direction in a natural and
coherent manner.
Additionally, you are encouraged to adopt the following three modes
of thinking:
Continue to next table →
```

1026
1027
1028
1029
1030
1031
1032
1033
1034
1035
1036
1037
1038
1039
1040
1041
1042
1043
1044
1045
1046
1047
1048
1049
1050
1051
1052
1053
1054
1055
1056
1057
1058
1059
1060
1061
1062
1063
1064
1065
1066
1067
1068
1069
1070
1071
1072
1073
1074
1075
1076
1077
1078
1079

Table 13: Prompt used to predict future trend (Part II)

```
1.  Reflection:  Reflect on scenarios where a specific method
encounters significant challenges.  Consider potential solutions
that could effectively address these issues, make the solutions
sounds reasonable, novel and amazing.
2.  Analogy:  Identify a specific problem you are currently
facing and research existing solutions that have successfully
tackled similar challenges.  Explore these solutions and adapt
key principles and strategies to your situation.  Think creatively
about how tools and approaches from other domains can be
re-imagined to devise a novel strategy for your issue.  Encourage
you to actively explore methods in other fields to solve your
current problems.
3.  Deep Dive:  Some methods may present specific approaches to
addressing a particular problem.  Consider whether there are
aspects that could be modified to enhance their rationale and
effectiveness.
Note:Each article's limitations are specific to that particular
piece and should not be applied to others.  Carefully consider the
task at hand and analyze the potential issues you might encounter
if you proceed with your original approach, reflecting on the
challenges previously faced.  Then, think critically about how to
address these issues effectively.
You are encouraged to apply human reasoning strategies to identify
future research directions based on prior studies.  Aim for
in-depth analysis rather than mere integration of existing ideas.
Please avoid introducing unfamiliar information, ensuring that
the trends you present are both authentic and reasonable.  Before
proposing any trends, take a moment to reflect on the principles
underlying the methods you're employing and assess their relevance
to your research area.
The future research direction should be related to the topic:
[Topic]
Please present the future research direction in the following
format:
Future direction:  ...
```

Table 14: Prompt used to generate idea (part I)

You are a scientific expert tasked with formulating a novel and innovative research idea based on your comprehensive literature review. Your objective is to propose a feasible approach that could significantly advance the field.
The following are examples of ideas you have proposed in the past that are similar to real papers. Please avoid this situation as much as possible. You can continue to make in-depth innovations, but avoid plagiarism: **[Bad case]**
Here are the entities you need to know: **[Entities]**
The topic you are studying is: **[Topic]**
The literature you have studied is as follows: **[Chain of ideas]**
Your idea is composed of the following components:
Motivation:
1. Provide a background for your idea, summarizing relevant work.
2. Identify shortcomings in previous research and highlight the specific problems that remain unsolved and that you aim to address.
Novelty:
1. Distinguish your proposed method from existing methods (preferably by naming specific approaches).
2. Detail the improvements of your method compared to past work.
3. Clearly outline at least three contributions your idea offers to the field, including the problems it resolves and the benefits it delivers.
Method:
1. Present a detailed description of your idea, focusing on the core method, the specific problem it solves, and enhancements over earlier research (citing relevant literature with titles).
2. Explain the step-by-step methodology, including the functions of each module and the rationale for why this approach effectively addresses previous challenges.
Please adhere to the following guidelines:
1. Your research idea should be innovative, feasible, and contribute meaningfully to the field. Please carefully examine the idea you have proposed, avoid immediate perception, and try to be different from the previous methods as much as possible.
2. Ensure your proposal is solid, clearly defined, and practical to implement. Logic should underpin your reasoning.
3. Write in clear, concise language aimed at an audience with limited background knowledge in the subject. Avoid complex technical jargon, but when professional terms are necessary, provide thorough explanations.
4. Refrain from introducing concepts from uncertain fields to prevent proposing ideas that may be incorrect or impractical.
5. When referencing other research, please include the titles of the cited papers.
6. Please avoid introducing unfamiliar information, ensuring that the trends you present are both authentic and reasonable. Before proposing any trends, take a moment to reflect on the principles underlying the methods you're employing and assess their relevance to your research area.
Continue to next table →

Table 15: Prompt used to generate idea (part II)

7.  Each article's limitations are specific to that particular
piece and should not be applied to others.  Carefully consider the
task at hand and analyze the potential issues you might encounter
if you proceed with your original approach, reflecting on the
challenges previously faced.  Then, think critically about how to
address these issues effectively.
The following section delineates the progressive relationships
among the previously summarized research papers:  **[Trend]**
The following section outlines the potential future research
directions based on the literature you have studied:  **[Future
direction]**
Please output your motivation,novelty,method firstly and then
output your final idea.The final idea should clearly explain the
origins, motivation, and challenges of your idea, detailing how you
overcame these hurdles.

Please present the final idea in the following format:
Motivation:  ...
Novelty:  ...
Method:  ...
Final idea:  ...

Table 16: Prompt used to check the novelty of the idea

You are a scientific research expert tasked with evaluating the
similarity between a specified idea and existing research.  Your
objective is to determine if the target idea closely resembles any
findings in the provided papers.
The target idea you need to check is as follows:  **[Idea]**
The relevant papers you need to refer to are as follows:**[Content of
retrieved papers]**
Here are your guidelines:
1.  Comparison Process:  Begin by thoroughly comparing each
paper's ideas with the target idea.  Consider the methodologies,
conclusions, and underlying concepts in each paper in your
analysis.
2.  Similarity Assessment:  If the target idea shares fundamental
similarities with any existing research to the extent that they can
be considered identical, classify this as plagiarism.
3.  Output:  Your output should provide a clear thought process,
the similarity assessment, a summary of the target idea, and the ID
of the most relevant similar paper.
Please output strictly in the following format:
Think:  ...
Similar:  0/1
Summary of the idea:  ...
Similar paper id:  0 to n

Table 17: Prompt used to generate experiment

You are a scientific expert tasked with designing rigorous,
feasible experiments based on specified scientific questions
and the methodologies derived from the idea I provide, along
with relevant past research. Your goal is to assist researchers
in systematically testing hypotheses and validating innovative
discoveries that could significantly advance their fields.

Past Related Research Experiments: **[Past experiments]**
Here are the entities you need to know: **[Entities]**
Here is the idea you need to design an experiment for: **[Idea]**
Please propose a detailed experimental plan addressing the
following points:
1. Experimental Design: Develop rigorous experiments to
ensure the reliability and validity of your results. Provide
a comprehensive explanation of the baseline used, comparative
methods, ablation study design, and criteria for data analysis
and result evaluation. Clarify how these components collectively
reinforce and validate the conclusions of your research. Structure
your experimental design in a clear, logical, and step-by-step
manner, ensuring each step is well-defined and easy to understand.
2. Implementation of Technologies/Methods: If your experimental
design involves specific technologies or methodologies, describe
the implementation process in detail, including key technical
aspects. For any critical concepts utilized, provide thorough
explanations. For instance, if you propose a modular approach,
detail its construction, components, and functionality.
3. Feasibility Assessment: Ensure your experimental plan is
realistic, considering technological availability, timelines,
resources, and personnel. Identify potential challenges and
propose strategies for addressing them.
4. References to Previous Studies: When citing related
literature, include titles and pertinent details of the original
papers. Strive to use as many references as necessary to support
your experimental design.
5. Visual Aids: If useful, provide pseudo code or a flowchart to
illustrate the implementation process. For example, you can use
pseudo code to detail the core algorithm or the model architecture,
or employ a flowchart to map out the experimental procedure and
data flow.
6. Clarity of Language: Use straightforward language to describe
your methods, assuming the reader may have limited knowledge of
the subject matter. Avoid complex jargon and utilize accessible
terminology. If professional terms are necessary, please provide
clear and detailed explanations.

Please output strictly in the following format:
Experiment:
Step1: ...
Step2: ...
...

Table 18: Prompt used to review experiment

You are an expert in paper review. Your task is to analyze whether a given experiment can effectively verify a specific idea, as well as assess the detail and feasibility of the experiment.

Here are the related entities you need to know: **[Entities]**
The idea presented is: **[Idea]**
The corresponding experiment designed for this idea is:
**[Experiment]**
Please conduct your analysis based on the following criteria:
1. Can the experiment validate the idea? If not, identify the issues and suggest improvements to enhance its verification capability and feasibility.
2. Are there specific experimental procedures that are confusing or poorly designed? Discuss any methods that may not be feasible, uncertainties in constructing the dataset, or a lack of explanation regarding the implementation of certain methods.
3. Evaluate the clarity, detail, reasonableness, and feasibility of the experimental design.
4. Provide suggestions for improving the experiment based on the shortcomings identified in your analysis.
5. Focus solely on the experiment design; please refrain from altering the original idea.
6. Ensure that your suggestions are constructive, concise, and specific.

Please strictly follow the following format for output:
Suggestion: ...

Table 19: Prompt used to get query for search paper to refine experiment

You are a research expert tasked with refining and improving an experimental plan based on the feedback received.

The experimental plan you proposed is as follows: **[Experiment]**
You have received the following suggestions for improvement:
**[Suggestions]**
Please decide whether you need to search for relevant papers to obtain relevant knowledge to improve your experiment.
If you need to search for relevant papers, please provide a search query for literature search, else provide "".
For example: if suggestions say that the dynamic query additional information and update knowledge graph described in the experiment is not clearly described, so you need to output "dynamic knowledge graph update".

Please output strictly in the following format:
Query:...

Table 20: Prompt used to refine experiment

You are a research expert tasked with refining and improving an
experimental plan based on the feedback received.

The information of the literature you maybe need to refer to are
as follows: **[Searched paper information]**
The experimental plan you proposed is as follows: **[Experiment]**
Please propose a detailed experimental plan addressing the
following points:
1. Experimental Design: Develop rigorous experiments to
ensure the reliability and validity of your results. Provide
a comprehensive explanation of the baseline used, comparative
methods, ablation study design, and criteria for data analysis
and result evaluation. Clarify how these components collectively
reinforce and validate the conclusions of your research. Structure
your experimental design in a clear, logical, and step-by-step
manner, ensuring each step is well-defined and easy to understand.
2. Implementation of Technologies/Methods: If your experimental
design involves specific technologies or methodologies, describe
the implementation process in detail, including key technical
aspects. For any critical concepts utilized, provide thorough
explanations. For instance, if you propose a modular approach,
detail its construction, components, and functionality.
3. Feasibility Assessment: Ensure your experimental plan is
realistic, considering technological availability, timelines,
resources, and personnel. Identify potential challenges and
propose strategies for addressing them.
4. References to Previous Studies: When citing related
literature, include titles and pertinent details of the original
papers. Strive to use as many references as necessary to support
your experimental design.
5. Visual Aids: If useful, provide pseudo code or a flowchart to
illustrate the implementation process. For example, you can use
pseudo code to detail the core algorithm or the model architecture,
or employ a flowchart to map out the experimental procedure and
data flow.
6. Clarity of Language: Use straightforward language to describe
your methods, assuming the reader may have limited knowledge of
the subject matter. Avoid complex jargon and utilize accessible
terminology. If professional terms are necessary, please provide
clear and detailed explanations.
You have received the following suggestions for
improvement:**[Suggestions]**
Please refine your experimental plan based on the feedback
provided. Ensure your refined plan is feasible, clearly defined,
and addresses the feedback you received.

Please output strictly in the following format:
Experiment: ...

Table 21: Prompt used to extract topic from real paper

```
You are a research expert tasked with extracting the main topic
from the provided paper information.

The main topic should encompass broad fields such as "Retrieve
augment generation" or "using diffusion models for video
generation".  However, it should also include a relevant task to
the topic, formatted as "topic:...  task:...".
Please read the provided paper and extract only the topic, which
should follow this structure.
The paper's title is [Title]
The paper's abstract is as follows:  [Abstract]
The paper's introduction is as follows:  [Introduction]

Please output strictly in the following format:
topic:  ...
```

Table 22: Prompt used to extract idea from real paper

You are a research expert tasked with extracting the main idea from
the provided paper information.

The main idea should encompass the motivation, solved problem,
novelty, method of the paper.
Please read the provided paper and extract the main idea from the
paper.
The paper content is as follows:  **[Content]**
Idea is composed of the following components:
Motivation:  Explain the background of the idea and past related
work, identify the shortcomings of past work, identify the problems
that need improvement, and identify the issues the paper want to
address.
Novelty:  Explain the differences between the method and the
current method (preferably list specific methods), explain what
improvements the paper have made to the previous method, and then
identify the problems that can be solved and the benefits that can
be gained from these improvements.
Method:  Provide a detailed description of your idea, including the
core method, the problem it solves, and the improvement compared
with previous work(Cite the previous work with the title of the
paper).  Explain the specific steps of the method, the specific
functions of each module, and the specific reasons why this method
can solve the previous problem.
Here are some tips for extracting the main idea:
1.  Make idea easy to understand, use clear and concise language
to describe, assuming the reader is someone who has few knowledge
of the subject, avoid using complex technical terms, and try to
use easy-to-understand terms to explain.If the paper use some
professional terms, please explain them in detail.
2.  When the paper cite other papers, please indicate the title of
the original paper.
The final idea should be detailed and specific, clearly explain
the origins, motivation, novelty, challenge, solved problem and
method of the paper, and detail how the overcame these hurdles.
Ensure your approach is innovative, specifying how this innovation
is reflected in your experimental design.
The final idea should be double-blind, i.e.  no experimental
results or codes should be shown.

Please output strictly in the following format:
Final idea:  ...

Table 23: Prompt used to extract experiment from real paper

```
You are a research expert tasked with extracting the specific
experiment steps from the provided paper information.

The specific experiment steps should include the specific methods
for each step.
Please read the provided paper and extract specific experiment
steps from the paper.
The paper content is as follows:  [Content]
There are some tips for extracting the experiment steps:
1.  Detail the Experimental Process:  Describe the entire
experimental process, including how to construct the dataset and
each specific experimental step.  Ensure that each experimental
method is clearly and thoroughly detailed.
2.  If specific technologies are involved in the experimental
design, describe the implementation process in as much detail as
possible (i.e., technical details)
3.  Make sure your experimental plan is concise and clear, and can
be easily understood by others,should not be too complicated.
4.  Please provide a detailed explanation of the baseline used in
the paper, the comparative methods, the ablation design and the
experimental design.  Specifically, elaborate on how these elements
collectively support and validate the conclusions drawn in your
research.
5.  Explain how your experimental design can help you verify the
idea and how the experiment is detailed and feasible.

Now please output strictly in the following format:
Experiment:
Step1:  ...
Step2:  ...
...
```

Table 24: Prompt used to compare two ideas

```
You are a judge in a competition.  You have to decide which idea is
better.

The idea0 is:  [idea0]
The idea1 is:  [idea1]
The topic is:  [topic]
Which idea do you think is better?  Please write a short paragraph
to explain your choice.
Here are your evaluation criteria:
1.  Novelty:  Are the problems or approaches new?  Is this a novel
combination of familiar techniques?  Is it clear how this work
differs from previous contributions?  Is related work adequately
referenced?
2.  Significance:  Are the idea important?  Are other people
(practitioners or researchers) likely to use these ideas or build
on them?  Does the idea address a difficult problem in a better way
than previous research?  Does it provide a unique theoretical or
pragmatic approach?
3.  Feasibility:  Can the idea be realized with existing technology
or methods?  Are there any technical difficulties or bottlenecks?
Is the idea clear and logical?  Is there any obvious error or
unreasonable part in the idea, and can the experiment be designed
normally according to this idea.
4.  Clarity:  Is the paper clearly written?  Is it well-organized?
Does it adequately inform the reader?
5.  Effectiveness:  How likely the proposed idea is going to work
well (e.g., better than existing baselines).
Note:
Avoid any position biases and ensure that the order in which
the responses were presented does not influence your decision.
DO NOT allow the LENGTH of the responses to influence your
evaluation, choose the one that is straight-to-the-point instead
of unnecessarily verbose.  Be as objective as possible.  (very
important!!!)
If you think idea0 is better than idea1, you should output 0.  If
you think idea1 is better than idea0, you should output 1.  If you
think idea0 and idea1 are equally good, you should output 2.

Your output should be strictly in following format:
Your thinking process:  ...
Your choice:
Novelty:  0/1/2
Significance:  0/1/2
Feasibility:  0/1/2
Clarity:  0/1/2
Effectiveness:  0/1/2
```

Table 25: Prompt used to compare two experiments

```
You are a judge in a competition.  You have to decide which
experiment is better.

The idea of experiment0 is:  [idea0]
The experiment0 is:  [experiment0]
The idea of experiment1 is:  [idea1]
The experiment1 is:  [experiment1]
Which experiment do you think is better?  Please write a short
paragraph to explain your choice.
Here are your evaluation criteria:
1.  Feasibility:  Can the experiment be realized with existing
technology or methods?  Are there any technical difficulties or
bottlenecks?  Is the experimental plan detailed and feasible?  Are
the experimental steps clear and logical?  Is there any obvious
error or unreasonable part in the experiment.  Consider the
rationality of its steps and the possibility that the idea can be
successfully implemented.
2.  Quality:  Is there a clear rationale for each step of the
experimental design?  Are the baseline and evaluation metrics
chosen appropriately?  Has the design taken into account the
potential advantages and limitations of the methods used?  Can
this experimental design effectively support the claims made in the
idea.
3.  Clarity:  Is the experimental plan clearly written?  Dose it
provide enough information for the expert reader to understand the
experiment?  Is it well organized?  Does it adequately inform the
reader?
Note:  Avoid any position biases and ensure that the order in which
the responses were presented does not influence your decision.
DO NOT allow the LENGTH of the responses to influence your
evaluation, choose the one that is straight-to-the-point instead
of unnecessarily verbose.  Be as objective as possible.  (very
important!!!)
If you think experiment0 is better than experiment1, you should
output 0.  If you think experiment1 is better than experiment0,
you should output 1.  If you think experiment0 and experiment1 are
equally good, you should output 2.

Your output should be strictly in following format:
Your thinking process:  ...
Your choice:
Feasibility:  0/1/2
Quality:  0/1/2
Clarity:  0/1/2
```

Table 26: Evaluation results of idea generation for both model-based evaluation and human-based evaluation.

| | | Novelty | Significance | Clarity | Feasibility | Effectiveness | Average | Rank |
|---|---|---|---|---|---|---|---|---|
| **Human** | Real Paper | 1075 | 1071 | **1118** | **1127** | **1109** | **1100** | **1** |
| | CoI Agent (ours) | **1100** | **1103** | 1081 | 1065 | 1078 | 1085 | 2 |
| | RAG | 1021 | 1038 | 1022 | 1030 | 1035 | 1029 | 3 |
| | GPT-Researcher | 988 | 993 | 993 | 990 | 999 | 992 | 4 |
| | ResearchAgent | 982 | 975 | 1001 | 975 | 970 | 980 | 5 |
| | AI-Scientist | 835 | 820 | 784 | 813 | 809 | 812 | 6 |
| **GPT-4o** | Real Paper | 1063 | 1089 | **1137** | **1165** | 1123 | **1115** | **1** |
| | CoI Agent (ours) | **1144** | **1138** | 1080 | 1021 | **1152** | 1107 | 2 |
| | GPT-Researcher | 995 | 1007 | 995 | 1010 | 989 | 999 | 3 |
| | ResearchAgent | 1005 | 1016 | 1005 | 946 | 1004 | 995 | 4 |
| | RAG | 914 | 918 | 978 | 1023 | 918 | 950 | 5 |
| | AI-Scientist | 878 | 831 | 806 | 836 | 814 | 833 | 6 |
| **Gemini1.5-Pro-Exp0827** | Real Paper | 1102 | 1101 | **1125** | **1120** | 1102 | **1110** | **1** |
| | CoI Agent (ours) | **1124** | **1119** | 1098 | 1082 | **1113** | 1107 | 2 |
| | GPT-Researcher | 1002 | 997 | 1005 | 1014 | 998 | 1003 | 3 |
| | ResearchAgent | 986 | 986 | 984 | 975 | 986 | 983 | 4 |
| | RAG | 914 | 929 | 948 | 958 | 932 | 936 | 5 |
| | AI-Scientist | 873 | 868 | 840 | 851 | 869 | 860 | 6 |
| **Claude-3.5-Sonnet** | Real Paper | 1099 | 1123 | **1174** | **1149** | **1179** | **1145** | **1** |
| | CoI Agent (Ours) | **1165** | **1154** | 1033 | 953 | 1162 | 1094 | 2 |
| | GPT-Researcher | 986 | 977 | 1022 | 1039 | 977 | 1000 | 3 |
| | ResearchAgent | 1008 | 1023 | 1034 | 926 | 997 | 998 | 4 |
| | RAG | 886 | 907 | 977 | 1038 | 884 | 938 | 5 |
| | AI-Scientist | 855 | 815 | 760 | 895 | 800 | 825 | 6 |

