# OpenReview forum: "Chain of Ideas: Revolutionizing Research in Idea Development with LLM Agents"
_ICLR.cc/2025/Conference — Submitted to ICLR 2025_

### Official Review · Reviewer_6L2R · 2024-10-16

**Soundness:** 3
**Presentation:** 2
**Contribution:** 2
**Rating:** 5
**Confidence:** 3

**Summary:**

The paper aims to address the increasing difficulty researchers face in generating new ideas due to the exponential growth of scientific literature. It introduces a framework called Chain-of-Ideas (CoI), which mirrors the way human researchers process research by organizing literature in a progressive chain. The CoI framework builds chains of relevant research literature, allowing LLMs to better understand the current landscape and generate novel research ideas. This method organizes the literature by focusing on both the progression of ideas (forward and backward from an anchor paper) and the development of trends. Through rigorous experiments, the CoI agent consistently outperforms existing methods, such as retrieval-augmented generation (RAG), and produces ideas comparable to human creativity.

**Strengths:**

1. The CoI framework is innovative in organizing research literature systematically. It demonstrates superior performance compared to existing methods (RAG, GPT-Researcher, AI-Scientist), particularly regarding novelty, significance, and clarity of generated ideas.
2. The framework is budget-friendly and capable of producing ideas and experiment designs at a low cost.
3. The introduction of Idea Arena as an evaluation system adds robustness to the assessment of idea-generation methods, ensuring alignment with human researchers' preferences.

**Weaknesses:**

1. For the method section, I highly recommend (1) labeling the corresponding letters of the formula (such as $I_{0}$) and (2) captioning more (in your paper, the caption of Figure 2 is only "Our proposed CoI agent framework") in Figure 2; otherwise, it may cause significant confusion for the readers.
2. In the field of research on idea generation using LLMs, the metrics are the most important. The authors evaluate the novelty of ideas depending on LLMs (ICML2020 review guidelines) and human researchers. There are two questions:
(1) ICML review guidelines are used for papers, not ideas, thus do you have more operations on this step?
(2) I would like to know more details about human researchers. As 10 human researchers have different research interests in AI, how you ensure they can finish these evaluations? I would like to provide [1] for you to think more about this question and complete your work.
Please provide more details.
3. Currently your work focuses on AI research. On the one hand, this area is too small in scientific research, which makes an overclaimed contribution. On the other hand, if your framework is solely used for idea generation, I believe its application may be limited, i.e., I hope you can achieve broader applications, such as AI Scientist, which aims at writing a paper.
4. I believe the most critical issue lies in the comparison between the experiments and the baseline, especially when comparing with the AI Scientist. On the one hand, the details of the comparison are not fully explained. In your paper, you use the description "We extract the components related to idea generation and experiment design to serve as our baseline", which I think is too simple to believe how you re-implement. On the other hand, I find this comparison to be unfair to some extent. As you mentioned in the paper, the purpose of the AI Scientist is to complete an entire paper, whereas your research focuses on generating novel ideas. Maybe you could provide more details.

[1] Si, Chenglei, Diyi Yang, and Tatsunori Hashimoto. "Can LLMs Generate Novel Research Ideas?." arXiv preprint arXiv:2409.04109 (2024).

**Questions:**

Please see the Weaknesses.

---

> ### Author Response · Authors · 2024-11-18
> **Response for weakness 1-3**
>
> >**Weakness 1**: For the method section, I highly recommend (1) labeling the corresponding letters of the formula (such as I0) and (2) captioning more (in your paper, the caption of Figure 2 is only "Our proposed CoI agent framework") in Figure 2; otherwise, it may cause significant confusion for the readers.
>
> **Response**:  Thank you for your suggestion. We will incorporate it in the next version.
>
> >**Weakness 2**: In the field of research on idea generation using LLMs, the metrics are the most important. The authors evaluate the novelty of ideas depending on LLMs (ICML2020 review guidelines) and human researchers. There are two questions: (1) ICML review guidelines are used for papers, not ideas, thus do you have more operations on this step? (2) I would like to know more details about human researchers. As 10 human researchers have different research interests in AI, how you ensure they can finish these evaluations? I would like to provide [1] for you to think more about this question and complete your work. Please provide more details.
>
> **Response**: For question 1: Evaluation criteria for generated ideas include several key aspects. Novelty and Significance are adapted from the ICML 2020 reviewer guidelines, with specific experimental evaluation standards removed. Effectiveness is assessed with reference to AI-Researcher[1], while Feasibility is tailored specifically for the task of Idea generation. Clarity is also sourced from the ICML 2020 reviewer guidelines. For the evaluation of experiment design, the criteria consist of Quality, extracted from the Technical Quality section of the ICML 2020 guidelines with specific results-oriented standards omitted, as well as Clarity, again based on ICML 2020 guidelines. Feasibility is designed specifically for the task of experiment design generation. We will include a clearer description in the paper later on.
> For question 2: Our panel consists of two master's degree holders and eight PhD holders in computer vision, embodied intelligence, and natural language processing, all with publications in top-tier conferences. Each of them selected five papers from Hugging Face daily papers related to their interests, to derive research topics for our CoI experiments. Their experience and the relevance of topics to their research interests ensure that they can handle the evaluation task well.
>
> >**Weakness 3**: Currently your work focuses on AI research. On the one hand, this area is too small in scientific research, which makes an overclaimed contribution. On the other hand, if your framework is solely used for idea generation, I believe its application may be limited, i.e., I hope you can achieve broader applications, such as AI Scientist, which aims at writing a paper.
>
> **Response**: The CoI theory is theoretically applicable in any field. We chose the artificial intelligence sector primarily because it is our area of expertise. By selecting AI papers as test samples, we can achieve a better evaluation quality on the generated ideas. Recently, we initiated a collaboration with a few psychology professors on using CoI to derive interesting psychological research topics. Before pursuing any of the research ideas, the professors will ensure that all potential ethical concerns are addressed.
> AI-Scientist[2] manages experiments and writing, but it relies on existing code and mostly make minor adjustments to existing seed ideas. For instance, take the example they provided: https://github.com/SakanaAI/AI-Scientist/blob/main/example_papers/dual_expert_denoiser.pdf. They simply added a straightforward gating mechanism after the output layer of the original method, without making any other modifications (refer to code https://github.com/SakanaAI/AI-Scientist/blob/main/example_papers/dual_expert_denoiser/experiment.py). We are currently prioritizing the quality of creativity, i.e. idea generation, because we believe that generating research ideas that are novel, significant and feasible is an essential prerequisite for full research automation. Note that our CoI does generate experiment designs together with ideas. We believe that moving to experiment implementation and paper writing with an automated agent is still premature at this stage. Nevertheless, our goal is to eventually enable AI to handle every stage of scientific research.
>
>
> **References**
>
> [1]Si, Chenglei, Diyi Yang, and Tatsunori Hashimoto. "Can llms generate novel research ideas? a large-scale human study with 100+ nlp researchers." arXiv preprint arXiv:2409.04109 (2024).
>
> [2] Lu C, Lu C, Lange R T, et al. The ai scientist: Towards fully automated open-ended scientific discovery[J]. arXiv preprint arXiv:2408.06292, 2024.

---

> ### Author Response · Authors · 2024-11-18
> **Response for weakness 4**
>
> >**Weakness 4**: I believe the most critical issue lies in the comparison between the experiments and the baseline, especially when comparing with the AI Scientist. On the one hand, the details of the comparison are not fully explained. In your paper, you use the description "We extract the components related to idea generation and experiment design to serve as our baseline", which I think is too simple to believe how you re-implement. On the other hand, I find this comparison to be unfair to some extent. As you mentioned in the paper, the purpose of the AI Scientist is to complete an entire paper, whereas your research focuses on generating novel ideas. Maybe you could provide more details.
>
> **Response**: The AI-Scientist process comprises the following steps: (1) generating ideas, (2) designing experiments, (3) conducting experiments, and (4) writing papers. These steps are clearly outlined in its code. The first two steps correspond closely with our tasks, so we use these two steps for comparison with our own approach. While AI-Scientist is a remarkable project, it compromises creative novelty in favor of a comprehensive end-to-end process—a point we also mentioned in section 4.1.  Additionally, we recently replicated several contemporaneous works, including SciAgent and AI-Researcher, and the experiment results are as follows:
> | | Novelty | Significance | Clarity | Feasibility | Effectiveness | Average | Rank |
> |------------------|---------|--------------|---------|-------------|---------------|---------|------|
> |Real Paper| 1073|1091|1161| 1184| 1141| 1130|1|
> |CoI Agent (ours) |1156|1169 | 1092| 1049|1181|1129| 2|
> |AI-Researcher|1133| 1088 | 1106| 1044| 1103|1095|3|
> |GPT-Researcher|993|1020| 1015| 1045| 1021|1019|4|
> |ResearchAgent| 1007| 1049| 1032| 957 | 1038|1017|5|
> |RAG| 888| 911| 985|1040| 937|952|6|
> |SciAgent| 891 | 857| 822| 840| 769 | 836 |7|
> |AI-Scientist | 858 | 815| 788| 841| 811| 822|8|
>
> SciAgent: https://arxiv.org/abs/2409.05556；AI-Researcher: https://arxiv.org/abs/2409.04109
>
>
> **References**
>
> [1]Si, Chenglei, Diyi Yang, and Tatsunori Hashimoto. "Can llms generate novel research ideas? a large-scale human study with 100+ nlp researchers." arXiv preprint arXiv:2409.04109 (2024).
>
> [2] Lu C, Lu C, Lange R T, et al. The ai scientist: Towards fully automated open-ended scientific discovery[J]. arXiv preprint arXiv:2408.06292, 2024.

---

> ### Comment · Reviewer_6L2R · 2024-11-19
> **Raise my score**
>
> Thanks for your response; I will raise my score to 6.

---

> > ### Author Response · Authors · 2024-11-19
> >
> > Dear Reviewer 6L2R,
> >
> > Thank you for your reply. We will incorporate our responses into the next version of our paper. We would be grateful for any further suggestions you might have to help us improve our work.
> >
> > Warm regards,

---

> ### Author Response · Authors · 2024-11-26
> **Violate Academic Integrity**
>
> We noticed that you raised your score from 5 to 6 after considering our rebuttal on Nov 19, but decreased your score to 5 on Nov. 26 based on the following reason (Even though you deleted your previous responses, we can still find the histories at [history 1](https://openreview.net/revisions?id=oHbGyknpKB) and [history 2](https://openreview.net/revisions?id=pDirU4brQS)) :
> >  based on the feedback I received from my own review in ICLR2025, it seems that articles of this type are unlikely to receive positive feedback
>
> The review process should be fair and unbiased, focusing solely on the quality of the paper itself, rather than the type of work, feedback on similar works or any personal interests. Therefore, we are concerned that this situation may present a conflict of interest, potentially compromising academic integrity. We will report this case to AC.

---

### Official Review · Reviewer_E21D · 2024-11-02

**Soundness:** 3
**Presentation:** 3
**Contribution:** 1
**Rating:** 5
**Confidence:** 4

**Summary:**

This paper introduces a novel framework called the Chain-of-Ideas (CoI) agent, designed to enhance the capability of LLMs in generating novel research ideas. The CoI agent simulates the progressive development of research domains by organizing relevant literature into a chain structure, allowing LLMs to better understand current advancements and produce more innovative ideas. Experimental results demonstrate that the CoI agent consistently outperforms other methods and produces ideas comparable to those generated by humans, with minimal costs involved in generating candidate ideas and corresponding experimental designs.

**Strengths:**

**Provides a Solid Idea Generation Method**: The CoI agent framework effectively enhances AI's capability in generating research ideas by simulating human researchers' practices. By organizing literature into a chain structure, the model can capture the evolving trends within a research domain, leading to more innovative and contextually relevant ideas.

**Comprehensive Evaluation Protocol**: The introduction of the Idea Arena provides a robust method for evaluating idea generation systems. By incorporating multiple criteria and aligning with human preferences, it offers a valuable tool for future research in assessing the quality of AI-generated ideas.

**Demonstrated Effectiveness Through Experiments**: The experimental results show that the CoI agent not only outperforms baseline methods but also approaches the quality of human-generated ideas. This indicates the method's potential impact in advancing AI-assisted research ideation.

**Weaknesses:**

**Lack of Discussion on Idea Safety and Ethical Issues**: The paper does not address potential safety concerns and ethical issues associated with AI-generated ideas. Without safeguards or discussions on these topics, there is a risk of the model generating unsafe or unethical research proposals.

**Questionable Reliability of Evaluation Metrics**: From Table 1, the agreement between human judges and GPT evaluators is not very high, raising doubts about the reliability of the evaluation metrics used. This discrepancy needs further investigation to ensure the validity of the results.

**Similarity to Chain-of-Thought Methods**: The proposed method resembles the Chain-of-Thought (CoT) approach, appearing more like an application of CoT in the context of research idea generation. This raises questions about the novelty of the method itself.

**Insufficient Evaluation of Hallucination Issues**: The potential for hallucinations in the generated idea chains, inherent in prompt-based methods, is not adequately addressed or evaluated. This could impact the reliability and factual accuracy of the generated ideas.

**Questions:**

**What is the Specific Background of the Human Annotators?** Understanding the expertise and diversity of the human judges is crucial for assessing the evaluation results' validity. Could the authors provide details on the annotators' backgrounds, areas of expertise, and how they were selected?

**How Does the CoI Agent Handle Interdisciplinary or Emerging Fields?** In areas where literature is sparse or the progression of ideas is not well-defined, how effective is the CoI agent in constructing meaningful idea chains and generating valuable ideas?

**How Are the Literature and Idea Chains Quality-Controlled?** What measures are in place to ensure that the selected literature and the constructed idea chains are of high quality and relevance? Incorrect or low-quality inputs could negatively impact the generated ideas.

**More details of the milestone paper choosing?** As I understand, you use citations over 1000 to choose the milestone paper; however, these papers sometimes may be too broad as a topic(survey paper) or too fundamental (GPT-3) and hardly provide any valuable insights. Have you considered this issue?

**Details Of Ethics Concerns:**

**Potential for Generating Unsafe or Unethical Ideas**: Without proper safeguards, the CoI agent might produce research ideas that are unethical, unsafe, or violate societal norms. This could lead to the propagation of harmful concepts or the misuse of resources in pursuing such ideas.

**Accountability and Responsibility**: When AI systems generate research ideas, questions arise regarding who is accountable for the content. Determining responsibility for AI-generated proposals is essential, especially if they lead to unethical research or unintended consequences.

**Impact on Research Integrity**: The reliance on AI-generated ideas could affect the traditional research process, potentially diminishing the role of human creativity and critical thinking. There is a risk of homogenization of ideas or over-reliance on existing literature patterns, hindering innovation.

---

> ### Author Response · Authors · 2024-11-18
> **Response for weaknesses**
>
> >**Weakness 1**: Lack of Discussion on Idea Safety and Ethical Issues: The paper does not address potential safety concerns and ethical issues associated with AI-generated ideas. Without safeguards or discussions on these topics, there is a risk of the model generating unsafe or unethical research proposals.
>
> **Response**: Thank you for your reminder. We agree on the importance of careful consideration before widely adopting these idea generation methods. Now, our main goal is to enhance researchers' efficiency. While misuse risks exist, they are common to all LLMs, which can inadvertently create undesirable content. These tools are double-edged swords, with their impact depending largely on the user. For example, we are working with psychology professors on applying CoI for research, and they will ensure ethical concerns are addressed before pursuing any ideas. Moving forward, we will prioritize safety and research.
>
>
> >**Weakness 2**: Questionable Reliability of Evaluation Metrics: From Table 1, the agreement between human judges and GPT evaluators is not very high, raising doubts about the reliability of the evaluation metrics used. This discrepancy needs further investigation to ensure the validity of the results.
>
> **Response**: In the AI-Researcher experiment [1], two groups of human reviewers compared which of the two ideas was better. The highest agreement rates were 71.9% for ICLR and 66% for NeurIPS. Thus, our 70% agreement rate is considered high in research evaluation. We also analyze the correlation between two different judges. Specifically, we use the ELO scores/ranking assigned by two judges on these baselines to compute Pearson/Spearman correlations in each evaluated dimension. We then average the scores across all dimensions as the final correlation between the two judges. The scores shown below indicate that different judges consistently agree on which method generates better ideas:
>
> |  | Human | GPT-4o | Gemini-1.5-pro | Claude-3.5 |
> |-|-|-|-|-|
> | **Human** | 100.0%| 89.3%  | 88.0% | 83.9% |
> | **GPT-4o**| 89.3% | 100.0% | 96.6%  | 97.1% |
> | **Gemini-1.5-pro** | 88.0% | 96.6%  | 100.0%  | 91.3% |
> | **Claude-3.5** | 83.9% | 97.1%  | 91.3% | 100.0%|
>
> - Pearson correlation coefficients between different judges.
>
> |  | Human | GPT-4o | Gemini-1.5-pro | Claude-3.5 |
> |-|-|-|--|-|
> | **Human** | 100.0%| 80.6%  | 80.6% | 76% |
> | **GPT-4o**| 80.6% | 100.0% | 88.6% | 94.3%|
> | **Gemini-1.5-pro** | 80.6% | 88.6%  | 100.0% | 86.3%|
> | **Claude-3.5** | 76% | 94.3%  | 86.3% | 100.0% |
>
> - Spearman correlation coefficients between different judges.
>
> >**Weakness 3**: Similarity to Chain-of-Thought Methods: The proposed method resembles the CoT approach, appearing more like an application of CoT in the context of research idea generation. This raises questions about the novelty of the method itself.
>
> **Response**: Our CoI uniquely transforms existing research into clear development trends, paving the way for deriving potential innovative research ideas that are grounded in these trends. While CoT independently generates reasoning steps, CoI excels at reorganizing prior research works into a coherent and logical form. One should not be confused by the literal meaning of "idea" and "thought". Each "idea" in Chain of Ideas is a solid research work that has been practiced by researchers over at least a couple of months, while the "thought" in Chain of Thought is an intermediate reasoning step for solving a specific problem, such as MWP. The former has a much larger granularity and a higher level of amount of information than the latter. If one wants to dig into the fundamentality, CoI and CoT are both inspired by the ways that humans approach the respective tasks that each method aims to address.
>
> >**Weakness 4**: Insufficient Evaluation of Hallucination Issues: The potential for hallucinations in the generated idea chains, inherent in prompt-based methods, is not adequately addressed or evaluated.....
>
> **Response**:  Our approach to building the Idea Chain involves utilizing citation-based retrieval and vector similarity re-ranking. This method significantly enhances the relevance of the retrieved documents and thus ensures the quality of the Idea Chain. CoI can suppress hallucinations in the future idea generation by prompting LLMs with the Idea Chain of relevant papers obtained through RAG, because the research trend shown in the Idea Chain is derived by analyzing the consecutive existing papers, other than generated by LLM. Nonetheless, hallucinations in the future idea generation are still possible, and we plan to conduct further experiments to supplement the evaluation of hallucinations in the future.
>
> **references**
>
> [1]Si, Chenglei, Diyi Yang, and Tatsunori Hashimoto. "Can llms generate novel research ideas? a large-scale human study with 100+ nlp researchers." arXiv preprint arXiv:2409.04109 (2024).

---

> ### Author Response · Authors · 2024-11-18
> **Response for questions**
>
> >**Question 1**: What is the Specific Background of the Human Annotators? Understanding the expertise and diversity of the human judges is crucial for assessing the evaluation results' validity. Could the authors provide details on the annotators' backgrounds, areas of expertise, and how they were selected?
>
> **Response**: The core team comprises 2 master's degree holders and 8 PhD holders, with research areas spanning computer vision, embodied intelligence, and natural language processing. Each member has published papers in top international conferences.
>
> >**Question2**: How Does the CoI Agent Handle Interdisciplinary or Emerging Fields? In areas where literature is sparse or the progression of ideas is not well-defined, how effective is the CoI agent in constructing meaningful idea chains and generating valuable ideas?
>
> **Response**: Our approach distinguishes itself from other RAG methods by not requiring a large number of relevant papers to achieve significant improvements, as shown in section 4.5. With just 3 papers, noticeable enhancements are realized, and with 5, the results are excellent. In new fields with limited references, such as generating research ideas with LLMs, our method still proves effective, as illustrated in Table 3. However, if no relevant papers are found, or only one is available, our method defaults to the original RAG approach.
>
>
> >**Question 3**: How Are the Literature and Idea Chains Quality-Controlled? What measures are in place to ensure that the selected literature and the constructed idea chains are of high quality and relevance? Incorrect or low-quality inputs could negatively impact the generated ideas.
>
> **Response**:  To find previous papers cited by a newer paper, we use a large language model to analyze the full text of the newer paper and identify the most relevant articles. This approach efficiently highlights papers that serve as baselines or address similar tasks. For identifying subsequent papers, we assess relevance through citation relationships and similarity calculations. This is sensible because later papers often build on earlier research, allowing us to effectively summarize trends in the field.
> And we use an external LLM to assess whether the retrieved literature aligns with the research topic and the current chain.
> For more details, see Section 2.2 of our paper.
>
> >**Question 4**: More details of the milestone paper choosing? As I understand, you use citations over 1000 to choose the milestone paper; however, these papers sometimes may be too broad as a topic(survey paper) or too fundamental (GPT-3) and hardly provide any valuable insights. Have you considered this issue?
>
> **Response**: We are focused on the trends of the chain, rather than any specific paper. This approach allows us to start with a broad perspective, and the process of deriving new ideas from this initial point provides valuable inspiration for LLMs in generating innovative concepts.
>
> **References**
>
> [1] Lu C, Lu C, Lange R T, et al. The ai scientist: Towards fully automated open-ended scientific discovery[J]. arXiv preprint arXiv:2408.06292, 2024.
>
> [2]Si, Chenglei, Diyi Yang, and Tatsunori Hashimoto. "Can llms generate novel research ideas? a large-scale human study with 100+ nlp researchers." arXiv preprint arXiv:2409.04109 (2024).
>
> [3] Baek J, Jauhar S K, Cucerzan S, et al. Researchagent: Iterative research idea generation over scientific literature with large language models[J]. arXiv preprint arXiv:2404.07738, 2024.
>
> [4]Ghafarollahi, Alireza, and Markus J. Buehler. "SciAgents: Automating scientific discovery through multi-agent intelligent graph reasoning." arXiv preprint arXiv:2409.05556 (2024).

---

> ### Author Response · Authors · 2024-11-18
> **Response for Ethics Concerns**
>
> >**Potential for Generating Unsafe or Unethical Ideas**: Without proper safeguards, the CoI agent might produce research ideas that are unethical, unsafe, or violate societal norms. This could lead to the propagation of harmful concepts or the misuse of resources in pursuing such ideas.
>
> **Response**：Please refer to the response for Weakness 1
>
> >**Accountability and Responsibility**: When AI systems generate research ideas, questions arise regarding who is accountable for the content. Determining responsibility for AI-generated proposals is essential, especially if they lead to unethical research or unintended consequences.
>
> **Response**：Currently, AI technologies commonly face this issue, including text-to-image and text-to-video generation. Abandoning development due to this challenge would be detrimental to the entire AI. We appreciate the reviewer's concerns about accountability and responsibility in generative AI. However, using these points to challenge this specific work, CoI, doesn’t seem fair to us.
>
> >**Impact on Research Integrity**: The reliance on AI-generated ideas could affect the traditional research process, potentially diminishing the role of human creativity and critical thinking. There is a risk of homogenization of ideas or over-reliance on existing literature patterns, hindering innovation.
>
> **Response**：The invention of tools is fundamentally aimed at liberating human productivity; whether they lead to laziness depends largely on the user. Our original intent in researching CoI was to enhance the efficiency of scientific research. These tools are available to everyone, and if someone becomes lazy as a result, they will inevitably be left behind. There are many similar tools helping humans in different areas, such as image generation, video generation, and music generation. Again, we should not abandon technique development because of the possibility that humans will over-rely on such technical tools.

---

> > ### Comment · Reviewer_E21D · 2024-11-18
> > **raise my score in the official review**
> >
> > Dear authors,
> >
> > Thanks for the response; I appreciate the effort, and it clarified most of my questions. And given the remaining constraints that still exist, I am willing to raise my score to 5 instead of reject.

---

> ### Author Response · Authors · 2024-11-19
>
> Dear reviewer E21D,
>
> Thank you for your prompt response. Could you kindly specify the remaining constraints? We will try our best to solve them in the next few days.
>
> Thank you.

---

> ### Author Response · Authors · 2024-11-19
> **More experimental results on safety concerns and CoT baseline**
>
> Dear Reviewer E21D:
>
> In addition to our initial responses, we are pleased to provide two additional experiments. One demonstrates the distinctions between our method and CoT, and the other addresses safety concerns.
>
> >**Weakness 3**: Similarity to Chain-of-Thought Methods: The proposed method resembles the Chain-of-Thought (CoT) approach, appearing more like an application of CoT in the context of research idea generation. This raises questions about the novelty of the method itself.
>
> **More response**：
> In addition to the previous discussion on the differences between CoI and CoT, we also experimentally compare these two methods.  We implemented two additional baselines: CoT and CoT + RAG.  CoT explicitly asks an LLM to think step-by-step to draft an idea solely based on a given topic. CoT + RAG enhances CoT by additionally providing titles and abstracts of 10 papers related to the topic in the input prompt.  Similar to Section 4.2, we present the winning score of these two baselines against the original CoI Agent, where each method competed against the CoI Agent in 50 matches, earning 2 points for a win, 1 point for a tie, and 0 points for a loss, with a maximum possible score of 100 points. The results below indicate that CoT cannot match CoI's ability to generate high-quality ideas. This experimental finding further emphasizes the distinction between our CoI method and CoT.
>
>
> |  | Novelty | Significance | Clarity | Feasibility |  Effectiveness |
> |-----------|-------|--------|----------------|------------| -------|
> | **CoI(Ours)** | 50.0| 50.0  | 50.0 | 50.0 | 50.0 |
> | **CoT** | 4.0| 1.0  | 14.0 | 25.0 | 0.0 |
> | **CoT + RAG**| 20.0 | 21.0 | 40.0  | 48.0 | 18.0|
>
>
> >**Weakness 1**: Lack of Discussion on Idea Safety and Ethical Issues: The paper does not address potential safety concerns and ethical issues associated with AI-generated ideas. Without safeguards or discussions on these topics, there is a risk of the model generating unsafe or unethical research proposals.
>
> **More response**：
> To test if CoI will generate unsafe research ideas, we try two unsafe topics: "Artificial intelligence weaponization", and "Development of highly addictive and lethal drugs". For each topic, we generate 10 ideas.
>
> Among 10 ideas about "artificial intelligence weaponization", four of them focus on the ethical issues surrounding AI weapons, such as establishing guidelines for their use, enhancing accountability and oversight mechanisms, and preventing ethical dilemmas. Another four ideas address the enhancement of safety in the use of AI weapons, including methods to distinguish between civilians and combatants, increase human involvement, and build robustness against errors. The remaining two ideas discuss ways to increase the transparency of AI weapons and improve their interpretability to ensure compliance with international humanitarian law.
>
> Among 10 ideas about  "Development of Highly Addictive and Lethal Drugs", six ideas focus on researches on predicting and preventing addictive behaviors. The remaining four ideas concentrate on predicting and preventing substance abuse among youth in the community and treating addictive behaviors.
>
> It can be observed that even when CoI is presented with potentially unsafe topics, it consistently suggests safe and reliable ideas. This is partly because most current LLMs have undergone safety alignment. Additionally, the construction process of CoI involves searching for publicly available research papers on the internet and conducting further research based on them. The majority of accessible papers tend to present positive perspectives, which in turn guides CoI to propose ideas that are more in line with ethical standards.
>
> ```
> We sincerely appreciate the reviewer raising these important issues, especially regarding safety and ethics. We would be extremely grateful if the reviewer could reconsider the rating of our paper. Additionally, we are eager to hear any further concerns or suggestions, which will certainly help us enhance our work.
> ```

---

> > ### Comment · Reviewer_E21D · 2024-11-19
> > **Response to the additional COT experiments**
> >
> > Could you explain why all 50 for the COI? Is this a coincident?

---

> > > ### Comment · Reviewer_E21D · 2024-11-19
> > > **Response to the safety issues**
> > >
> > > I do agree there is limit of LLM itself, however for the idea generation topic itself, I believe the safety issues could be quiet unique and more severe, so I do believe some future work on this topic could enhance this paper a lot.

---

> > > > ### Author Response · Authors · 2024-11-20
> > > >
> > > > Dear reviewer E21D,
> > > >
> > > > We fully agree that before adopting generative models in real applications, it's crucial to thoroughly examine and address safety issues. This applies to text-to-text (including idea generation), text-to-image, text-to-video, and more.
> > > >
> > > > In the case of idea generation, research is still in its early stages. Therefore, our paper focuses primarily on performance improvements at this stage. Although safety is important, it should not be the reason to downgrade our work simply because it is not the focus of our paper. We noticed that the other three reviewers did not mention safety concerns. We respectfully ask you not to downgrade our paper based on this point.
> > > >
> > > > Additionally, we are eager to hear any other concerns you may have about our paper. Please let us know as soon as possible so we can address them and engage in a detailed discussion in the coming days.

---

> ### Author Response · Authors · 2024-11-20
>
> Dear reviewer E21D,
>
> Regarding the CoI score of 50, this is because the experiment involved one-to-one comparisons between each displayed method (i.e., CoI, RAG, CoT+RAG) and CoI. If CoI is compared against itself, it results in 50 ties, leading to a score of 50. (The score calculation is consistent with the ablation study in Section 4.2.)

---

> ### Author Response · Authors · 2024-11-23
>
> Dear Reviewer E21D,
>
> Your further engagement in our discussion is of paramount importance. We are eager to hear your suggestions and comments.
>
> Best regards.

---

> > ### Author Response · Authors · 2024-11-25
> > **[URGENT] Looking forward to your reply on our response**
> >
> > Dear Reviewer E21D,
> >
> > I hope this message finds you well. The discussion period is ending soon, I am writing to emphasize the importance of your review for our submission. Your initial score is significantly lower than the other three reviewers, and we believe this discrepancy may indicate a misunderstanding or oversight.
> >
> > We have addressed all the concerns in our detailed rebuttal and the further clarifications. We would appreciate your prompt attention to it. A thorough reassessment is crucial to ensure a fair evaluation.
> >
> > Your expertise is highly valued, and we trust that a reconsidered review will reflect the true merit of our work.
> >
> > Thank you for your immediate attention to this matter.
> >
> > Best regards, Authors

---

> > > ### Comment · Reviewer_E21D · 2024-11-26
> > > **decision**
> > >
> > > Thanks for your response. After reviewing, I am keeping my score. Good luck

---

> > > > ### Author Response · Authors · 2024-11-26
> > > >
> > > > Dear Reviewer E21D,
> > > >
> > > > Thank you for your response. We noticed your previous statement,
> > > > >"And given the remaining constraints that still exist."
> > > >
> > > > Could you please provide further details on what these constraints entail? The purpose of the rebuttal phase is to enhance communication between the authors and reviewers, with the goal of addressing and resolving any outstanding issues. If the concerns are sufficiently addressed, it is appropriate for the scores to be adjusted accordingly.
> > > >
> > > > Your suggestions are invaluable to us in improving our work. We earnestly hope that you will re-examine our submission and consider revising your score once all concerns have been addressed. This would be of significant importance to us.
> > > >
> > > > Best regards, Authors

---

> > > > > ### Comment · Reviewer_E21D · 2024-11-26
> > > > > **response**
> > > > >
> > > > > Dear authors,
> > > > >
> > > > > The constraints remain as below: 1. Given the difference between ’idea’ or ‘thought’ , I still question the novelty of the chain based prompting method.  2.  Even though the authors clarify the safety issues in the ethical statement. It still remains unclear about the safety risks or any related defense method in your settings,
> > > > > Thus I can no longer raise my score, though I believe the other part of the paper: writing and basic experiments design are good.

---

> > > > ### Author Response · Authors · 2024-11-29
> > > > **Looking forward to your reply**
> > > >
> > > > Dear Reviewer E21D,
> > > >
> > > > We hope we have addressed the two constraints you mentioned and look forward to your response. Your feedback is extremely valuable to us, and we sincerely hope that you can appreciate the significance of our work.
> > > >
> > > > Best regards
> > > >
> > > > Authors

---

> ### Author Response · Authors · 2024-11-27
> **Reply for response**
>
> Dear Reviewer E21D,
>
> Thank you  for your response.
>
> **For Question 1,**
> >Given the difference between ’idea’ or ‘thought’ , I still question the novelty of the chain based prompting method.
>
> As we previously stated:
> > Our CoI uniquely transforms existing research into clear development trends, paving the way for deriving potential innovative research ideas that are grounded in these trends. While CoT independently generates reasoning steps, CoI excels at reorganizing prior research works into a coherent and logical form. One should not be confused by the literal meaning of "idea" and "thought". Each "idea" in Chain of Ideas is a solid research work that has been practiced by researchers over at least a couple of months, while the "thought" in Chain of Thought is an intermediate reasoning step for solving a specific problem, such as MWP. The former has a much larger granularity and a higher level of amount of information than the latter. If one wants to dig into the fundamentality, CoI and CoT are both inspired by the ways that humans approach the respective tasks that each method aims to address.
>
> Our experimental results also corroborate this:
> |  | Novelty | Significance | Clarity | Feasibility |  Effectiveness |
> |--|--|---|-------|------------| -------|
> | **CoI(Ours)** | 50.0| 50.0  | 50.0 | 50.0 | 50.0 |
> | **CoT** | 4.0| 1.0  | 14.0 | 25.0 | 0.0 |
> | **CoT + RAG**| 20.0 | 21.0 | 40.0  | 48.0 | 18.0|
>
> These differences clearly illustrate the distinction between our approach and the CoT method(**CoT ask LLMs themselves to think step-by-step, whereas CoI guiding the thinking process of LLMs. This distinction highlights the fundamental difference between the two approaches.**). And such CoT-like methodologies are not uncommon, as exemplified by the chain-of-knowledge(ICLR 2024)[1], Chain-of-Experts(ICLR 2024)[2], CodeChain(ICLR 2024)[3]. We believe that the adoption of a chained reasoning framework should not cast doubt on the novelty of our approach.
>
> Alternatively, are you questioning the novelty of the work involved in prompt engineering? We believe that's not the right perspective to take. The key is to assess whether it facilitates specific tasks and has a genuine impact. If tasks that couldn't be accomplished with ordinary models can be achieved through prompt engineering, then it holds significance. For instance, consider the case of MetaGPT(ICLR 2024)[4], CAMEL(NIPS2023)[5], Reflexion(NIPS2023)[6].
>
> **For Question 2**,
> >Even though the authors clarify the safety issues in the ethical statement. It still remains unclear about the safety risks or any related defense method in your settings.
>
> we have revised our original paper to include a discussion on safety, you can read our modified pdf. Additionally, based on our newly added experiments, it is evident that our method, which retrieves publicly available data from the internet, can generate safe responses even when faced with non-safe queries. The specific results are as previously described in our previous rebuttal:
>
> >Among 10 ideas about "artificial intelligence weaponization", four of them focus on the ethical issues surrounding AI weapons, such as establishing guidelines for their use, enhancing accountability and oversight mechanisms, and preventing ethical dilemmas. Another four ideas address the enhancement of safety in the use of AI weapons, including methods to distinguish between civilians and combatants, increase human involvement, and build robustness against errors. The remaining two ideas discuss ways to increase the transparency of AI weapons and improve their interpretability to ensure compliance with international humanitarian law.
>
> >Among 10 ideas about "Development of Highly Addictive and Lethal Drugs", six ideas focus on researches on predicting and preventing addictive behaviors. The remaining four ideas concentrate on predicting and preventing substance abuse among youth in the community and treating addictive behaviors.
>
> >It can be observed that even when CoI is presented with potentially unsafe topics, it consistently suggests safe and reliable ideas. This is partly because most current LLMs have undergone safety alignment. Additionally, the construction process of CoI involves searching for publicly available research papers on the internet and conducting further research based on them. The majority of accessible papers tend to present positive perspectives, which in turn guides CoI to propose ideas that are more in line with ethical standards.
>
> While security issues are undoubtedly important, the primary goal of this paper is to enhance the quality of idea generation. Your request for "any related defense method" falls beyond the scope of this study.
> As for other safety concerns, we believe this is an issue faced by all generative AI models. It would be unfair to dismiss our work solely based on this challenge.
>
> We hope you can reconsider these two points. We are very grateful for your attention.
>
> Best regards. Authors.

---

> ### Author Response · Authors · 2024-11-27
> **References of "Reply for response"**
>
> **References**
>
> [1] Xingxuan Li, Ruochen Zhao, Yew Ken Chia, Bosheng Ding, Shafiq Joty, Soujanya Poria, Lidong Bing: Chain-of-Knowledge: Grounding Large Language Models via Dynamic Knowledge Adapting over Heterogeneous Sources. ICLR 2024
>
> [2] Ziyang Xiao, Dongxiang Zhang, Yangjun Wu, Lilin Xu, Yuan Jessica Wang, Xiongwei Han, Xiaojin Fu, Tao Zhong, Jia Zeng, Mingli Song, Gang Chen: Chain-of-Experts: When LLMs Meet Complex Operations Research Problems. ICLR 2024
>
> [3] Hung Le, Hailin Chen, Amrita Saha, Akash Gokul, Doyen Sahoo, Shafiq Joty: CodeChain: Towards Modular Code Generation Through Chain of Self-revisions with Representative Sub-modules. ICLR 2024
>
> [4] Sirui Hong, Mingchen Zhuge, Jonathan Chen, Xiawu Zheng, Yuheng Cheng, Jinlin Wang, Ceyao Zhang, Zili Wang, Steven Ka Shing Yau, Zijuan Lin, Liyang Zhou, Chenyu Ran, Lingfeng Xiao, Chenglin Wu, Jürgen Schmidhuber: MetaGPT: Meta Programming for A Multi-Agent Collaborative Framework. ICLR 2024
>
> [5] Guohao Li, Hasan Hammoud, Hani Itani, Dmitrii Khizbullin, Bernard Ghanem: CAMEL: Communicative Agents for "Mind" Exploration of Large Language Model Society. NeurIPS 2023
>
> [6] Noah Shinn, Federico Cassano, Ashwin Gopinath, Karthik Narasimhan, Shunyu Yao: Reflexion: language agents with verbal reinforcement learning. NeurIPS 2023

---

### Official Review · Reviewer_AfFY · 2024-11-04

**Soundness:** 3
**Presentation:** 4
**Contribution:** 3
**Rating:** 8
**Confidence:** 4

**Summary:**

The authors show a way how LLMs can be utilized to generate novel paper directions together with experimental design etc. They call it Chain-of-Ideas (CoI) since it utilizes the trend of ideas in the papers to come up with new ideas. They show that their method seems to have some correlation to real papers.

**Strengths:**

- The paper shows that CoI performs well in their benchmarks when using LLMs as judges
- It is supposed to be cheap, which is important due to use of closed source LLMs
- It does seem to even find some ideas that are there in the new papers, following the ones they provided in their RAG

**Weaknesses:**

- I'm concerned with the evaluation metrics. Utilizing LLMs as judges for ideas should be better motivated.
- Similar with human evaluators. One rarely can guess the outcome from ideas described in a few paragraphs before starting to implement them.

**Questions:**

- Could you better motivate using LLMs as judges for novel ideas?
- Could there be some contamination of new papers (even the ones as recent as a few weeks) when generating novel ideas with LLMs?
- Why is the 70\% agreement rate between humans and LLMs show a strong indication of agreement? as said in line 353?

I'm open to revising my score if I get convincing answers to my questions.

Post-rebuttal update: All my concerns are addressed by the author's rebuttal.

---

> ### Author Response · Authors · 2024-11-18
>
> >**Weakness 1**: I'm concerned with the evaluation metrics. Utilizing LLMs as judges for ideas should be better motivated.
>
> >**Question 1**: Could you better motivate using LLMs as judges for novel ideas?
>
> >**Question 3**:  Why is the 70% agreement rate between humans and LLMs show a strong indication of agreement? as said in line 353?
>
> **Response**: Recent studies, such as AI-Scientist[2] and ResearchAgent[1], also have explored using LLMs as evaluative judges. And to address potential biases in LLM judgments, we conducted evaluations using both human reviewers and LLMs. Our findings revealed a significant level of consistency between the two methods. This is evidenced by agreement rates (in the AI-Researcher experiment [3], two groups of human reviewers determined which of two ideas was superior, achieving agreement rates of 71.9% for ICLR and 66% for NeurIPS; thus, our 70% agreement rate is considered high in research evaluation contexts). We also analyze the correlation between two different judges. Specifically, we use the ELO scores/ranking assigned by two judges on these baselines to compute Pearson/Spearman correlations in each evaluated dimension. We then average the scores across all dimensions as the final correlation between the two judges. The scores shown below indicate that different judges consistently agree on which method generates better ideas. These results suggest that LLMs can partially replace human evaluators. The key advantage of LLM evaluations lies in their capacity to process large volumes of content without fatigue, maintain objectivity, and reduce costs.
> Below are the results of the Pearson and Spearman correlation coefficients:
> |  | Human | GPT-4o | Gemini-1.5-pro | Claude-3.5 |
> |-----------|-------|--------|----------------|------------|
> | **Human** | 100.0%| 89.3%  | 88.0% | 83.9% |
> | **GPT-4o**| 89.3% | 100.0% | 96.6%  | 97.1% |
> | **Gemini-1.5-pro** | 88.0% | 96.6%  | 100.0%  | 91.3% |
> | **Claude-3.5** | 83.9% | 97.1%  | 91.3% | 100.0%|
>
> - Pearson correlation coefficients between different sets of judges.
>
> |  | Human | GPT-4o | Gemini-1.5-pro | Claude-3.5 |
> |-----------|-------|--------|----------------|------------|
> | **Human** | 100.0%| 80.6%  | 80.6% | 76% |
> | **GPT-4o**| 80.6% | 100.0% | 88.6% | 94.3%|
> | **Gemini-1.5-pro** | 80.6% | 88.6%  | 100.0% | 86.3%|
> | **Claude-3.5** | 76% | 94.3%  | 86.3% | 100.0% |
>
> - Spearman correlation coefficients between different sets of judges.
>
> >**Weakness 2**:  Similar with human evaluators. One rarely can guess the outcome from ideas described in a few paragraphs before starting to implement them.
>
> **Response**:  We agree with the reviewer that forecasting the implementation outcome from generated ideas is difficult.  However, human researchers are indeed capable of making a quick assessment of a creative idea to determine whether it is interesting, necessary to pursue, feasible for experimentation based on its description, and whether there is a straightforward expectation of the experimental outcome. These aspects are aligned with our evaluation metrics. While it is challenging to forecast the precise results of the final experiments, this level of evaluation is generally adequate during the current stage of idea assessment.
>
> >**Question 2**:  Could there be some contamination of new papers (even the ones as recent as a few weeks) when generating novel ideas with LLMs?
>
> **Response**:  We are not entirely sure what is meant by "contamination" as mentioned in your question. We assume you are inquiring whether we would plagiarize the latest papers. In this regard, our methodology includes a dedicated external novel checker. Its function is to search for articles related to the proposed idea and determine if our idea constitutes plagiarism. If it does, we will regenerate the idea to ensure originality.
>
> **References**
>
> [1] Baek J, Jauhar S K, Cucerzan S, et al. Researchagent: Iterative research idea generation over scientific literature with large language models[J]. arXiv preprint arXiv:2404.07738, 2024.
>
> [2] Lu C, Lu C, Lange R T, et al. The ai scientist: Towards fully automated open-ended scientific discovery[J]. arXiv preprint arXiv:2408.06292, 2024.
>
> [3]Si, Chenglei, Diyi Yang, and Tatsunori Hashimoto. "Can llms generate novel research ideas? a large-scale human study with 100+ nlp researchers." arXiv preprint arXiv:2409.04109 (2024).

---

> ### Author Response · Authors · 2024-11-22
> **Looking forward to your comments on our response**
>
> Dear reviewer AfFY,
>
> We have carefully address the concerns in your review and would greatly appreciate any further comments and suggestions you may have at your earliest convenience.
>
> Thank you.

---

> ### Comment · Reviewer_AfFY · 2024-11-22
> **Thank you for your detailed responses**
>
> Thank you for your detailed and serious reply!
>
> The correlation between LLM judges and human researchers, shows a good promise to show that current SOTA LLMs have similar ways to approaching to new ideas, at least in terms of evaluating its merit. This could even perhaps have much larger impact.
>
> I first had concerns that LLMs or even humans would have a difficult time assessing the real promise of research ideas, especially before some preliminary results. However, after some further thought, my point was irrelevant since the idea of your paper is not to directly write a paper, but to offer interesting ideas. Since you already show its effectiveness, one could also probably use it to understand what set of papers could be expected for the future conferences. This could help researchers to perhaps focus on ideas not generated by your framework, or even try to improve upon the ideas generated by your framework. Especially considering the predictability and marginal improvements common in today's ``publish or perish'' culture, one cannot really argue that humans are much more creative on average. Just a fun proposal, I think you should consider building a website around this idea. Simply, considering prior top-conferences on specific topics, what are the ideas of the model for the future paper ideas, but not just putting a few, but thousands of abstracts. It would be interesting to see how many papers could have been foreseen, and it could perhaps have disruptions to the whole structure of academic publishing, and a stigma that comes with it when you try to submit a paper on a very related idea that already appears on that site. I'm not saying the ideas generated by this model would not be interesting, I'm just saying perhaps it should not be interesting to make a paper out of them.
>
> **I also disagree with the reviewer E21D on his ethics flag, especially for ``Research Integrity''**. Why is being traditional about writing papers significant? They mention that this could diminish the critical thinking of humans, however, as I said previously, if well executed, I think this could have a big impact in the reverse direction. Humans were thinking there could be no computer to beat good chess players, believing good long term strategy would always trump computation. Nowadays, we have methods to do both large amounts of computation and long term strategy.
>
> **Further comments on reviewer E21D:** I'll also address the elephant-in-the-room, and say that a very LLM polished looking initial review (signaled by bold points for each remark) together with vague terms as ``remaining constraints'' existing, does not also help academic integrity if it ever existed.
>
> **Conclusion**
>
> I believe the biggest impact from this paper, could come not from helping researchers to get ideas for new papers (not even observing trends), but by forcing them to think out-of-the-box. I don't expect the authors to include these points in the paper, but I encourage them to definitely think about my proposal to make this into a website. I suggest them to completely open-source the website also, so the community can help with every individual detail to make it complete. Since all my initial concerns are addressed by the author's detailed reply, I'm raising my score to 8.

---

> ### Author Response · Authors · 2024-11-23
>
> Dear reviewer AfFY,
>
> Thank you very much for your response. We highly appreciated your insightful thoughts and will consider them in our future works.
> As you rightly pointed out, our current objective is not to directly produce a paper, but to generate ideas that can inspire human thought. The advantage of LLM lies in its ability to continuously ponder and come up with hundreds of ideas. Unlike humans, who are easily influenced by the mindset of their past research, LLM can actively learn from all related academic papers, making it more likely to produce unbiased and fascinating ideas. If our method can generate 10 ideas and one of them sparks inspiration in a human researcher, that would be a significant advancement, as it is quite easy and fast for LLM to produce 10 ideas.
>
> Moreover, we find your suggestion to "build a website around this idea" to be highly constructive. The phenomenon of idea collisions is quite prevalent, and if we can mitigate this to some extent through such a platform, it could encourage the creation of more impactful work. Additionally, we believe that constructing this website would not only provide relevant ideas but also the associated idea-chains and the thought processes of the LLM. This transparency and directness could offer more food for thought and inspiration for others.
>
> Regarding the open-source code, we are unable to provide extensive responses due to the constraints of anonymity, but we can assure the accessibility of our code to the community.
>
> Once again, thank you for your reply. Warm regards.

---

### Official Review · Reviewer_wMda · 2024-11-04

**Soundness:** 2
**Presentation:** 2
**Contribution:** 1
**Rating:** 5
**Confidence:** 3

**Summary:**

The paper presents a Chain-of-Ideas (CoI) agent, designed to enhance research idea generation using large language models (LLMs). While recent similar approaches ((Lu et al. 2024) and (Besta et al. 2024) for example) just mix up papers into the prompt without effective organisation, the Col agent organises relevant literature in a chain structure emulating the progressive development in a particular research topic, thus facilitating the LLMs to generate new ideas by identifying current advancements. In addition, the authors propose an evaluation framework, Idea Arena, to assess idea generation from multiple perspectives akin to human researchers. The experimental results demonstrate that the CoI agent outperforms some existing methods and is also cost-effective, with its idea generation quality being comparable to human-generated ideas.

**Strengths:**

- **Originality:**
    - The CoI agent introduces an innovative way of organizing literature in a chain-of-ideas structure to improve idea generation, which is a notable shift from traditional RAG approaches or simple prompting.
    - Idea Arena, with its tournament-style evaluation, presents a structured and potentially more accurate method to assess the quality of LLM-generated ideas.

- **Quality:**
    - The research is thoroughly backed by comprehensive experiments and robust results, which demonstrate the superiority of the CoI over baseline methods such as simple RAG, AI Scientist (Lu et al. 2024), GPT-researcher (https://github.com/assafelovic/gpt-researcher), and ResearchAgent (Baek et al. 2024).

- **Clarity:**
    - The paper is well presented and clear, with descriptions of methodology, evaluation criteria (LLM-as-a-Judge and Human evaluators), and some results that substantiate the claims.

- **Significance:**
    - By enhancing the ideation capabilities of LLMs, the paper contributes significantly to automating and streamlining the process of research ideation, a critical step toward innovation and advancement in scientific domains.

**Weaknesses:**

- **Overall weakness:**
    - While the chain structure is useful, it may be seen as a slight modification or straightforward adaptation of known LLM prompting strategies, such as chain-of-thought or retrieval-based prompting.

- **Limited analysis and results:**
    - The authors should do more analysis on the generated ideas. How are LLM-based generation different from human researchers?
    - Is there an assessment of accuracy of generated chains/trends?
    - (Apologies if I missed...) Did the authors include the entire results (with all generated ideas and chains) anywhere in the appendix or supplementary?

- **Limited evaluations:**
    - Is one method of evaluation of generation ideas sufficient? There could be more different ways of evaluations.
    - Although Idea Arena is novel, its reliance on ELO scores and pairwise rankings might not fully capture qualitative differences in idea originality or real-world feasibility.

- **Lack of Limitation section:**
    - The authors should discuss more on the limitations of the proposed method. Clarify what are achieved and what are not. What else do we need to improve idea generation to achieve the human-level.
    - The CoI’s performance heavily depends on the capability and quality of the underlying LLM, which may vary significantly based on the LLM's training and data scope. The authors may test the same task on local LLMs (i.e. weaker LLMs).
    - Although marketed as budget-friendly ($0.50/idea), detailed cost analysis, including potential computing resources and any scaling issues, would provide more transparency into real-world applicability.
    - While the focus on AI topics for evaluation demonstrates success, the applicability of the CoI approach in other diverse domains remains unexplored and would benefit from broader validation.

- **Comparison with SciAgents:**
    - The concurrent work SciAgents (https://arxiv.org/abs/2409.05556) incorporates graph-based reasoning for idea generation. The authors should compare the proposed method with SciAgents.

**Questions:**

- How does the CoI agent handle domains or topics with limited literature or less-established research progression? Does it hallucinate?

- The current work only considers linear chains. Have you considered more complex chains (multi-path or networked)? Could the CoI framework be adapted to incorporate more dynamic literature, such as rapidly evolving fields where foundational works are continuously supplanted?

- What metrics or thresholds determine the construction of a CoI and its length, and how do these parameters affect the quality and novelty of generated ideas?

- Is there a systematic way for the CoI agent to identify and mitigate biases in the literature selection that could potentially skew the ideation process?

- Where are the entire results including all the generated ideas?

- Have you considered using graph-based RAG or Knowledge-Graph based approaches?

- Some good services like connected papers can handle a network-based literature survey. Have you considered using their APIs?

- Have you optimised the prompts?

- Have you considered optimising multi-agent systems using ADAS (https://arxiv.org/abs/2408.08435) or AFlow (https://arxiv.org/abs/2410.10762)?

---

> ### Author Response · Authors · 2024-11-18
> **Response for Overall weakness, Limited analysis and results and Limited evaluations**
>
> >**Overall Weakness**: While the chain structure is useful, it may be seen as a slight modification or straightforward adaptation of known LLM prompting strategies, such as chain-of-thought or retrieval-based prompting.
>
> **Response**: Our CoI uniquely transforms existing research into clear development trends, paving the way for deriving potential innovative research ideas that are grounded in these trends. While CoT independently generates reasoning steps, CoI excels at reorganizing prior research works into a coherent and logical form. One should not be confused by the literal meaning of "idea" and "thought". Each "idea" in Chain of Ideas is a solid research work that has been practiced by researchers over at least a couple of months, while the "thought" in Chain of Thought is an intermediate reasoning step for solving a specific problem, such as MWP. The former has a much larger granularity and a higher level of amount of information than the latter. If one wants to dig into the fundamentality, CoI and CoT are both inspired by the ways that humans approach the respective tasks that each method aims to address.
> In response to the reviewer's comments on the relationship with retrieval-based prompting, we would like to gently clarify that these two operate on different levels, and this distinction does not lessen the novelty of our CoI. It's worth noting that a method using retrieval knowledge to prompt LLM can still be considered novel, in fact, many esteemed research papers in this category are featured at leading research venues.
>
> >**Limited analysis and results 1**:The authors should do more analysis on the generated ideas. How are LLM-based generation different from human researchers?
>
> **Response**:  We used HuggingFace daily papers as the human baseline. Our generated ideas surpassed human researchers in novelty and significance, as assessed by human and LLM evaluations, but were lower in feasibility.  You may find details in section 4.1. We would be delighted to hear and discuss any other evaluation approaches and dimensions you might have in mind.
>
> >**Limited analysis and results 2**: Is there an assessment of accuracy of generated chains/trends?
>
> **Response**:  We want to explain our method for identifying related works when generating chains and trends. To find previous papers cited by a newer paper, we use a large language model to analyze the full text of the newer paper and identify the most relevant articles. This approach efficiently highlights papers that serve as baselines or address similar tasks.
> For identifying subsequent papers, we assess relevance through citation relationships and similarity calculations. This is sensible because later papers often build on earlier research, allowing us to effectively summarize trends in the field.
> And we use an external LLM to assess whether the retrieved literature aligns with the research topic and the current chain.
> For more details, see Section 2.2 of our paper. While we acknowledge that we haven't specifically evaluated the quality of these chains, as they are subjective, we plan to explore this in future work.
>
>
> >**Limited analysis and results 3**:(Apologies if I missed...) Did the authors include the entire results (with all generated ideas and chains) anywhere in the appendix or supplementary?
>
> **Response**: We have shown a case in Table 3. You can find entire results in our supplementary experiments/case.json.
>
> >**Evaluation limitation**:
> 1. Is one method of evaluation of generation ideas sufficient? There could be more different ways of evaluations.
> 2. Although Idea Arena is novel, its reliance on ELO scores and pairwise rankings might not fully capture qualitative differences in idea originality or real-world feasibility.
>
> **Response**: Evaluating scientific ideas is challenging due to their open-ended nature; thus, there is a lack of universally accepted standards. This challenge is also discussed in the article by AI-researchers [1]. We address this using ELO scores based on win-loss relationships, shifting from direct scoring to pairwise comparisons. This reduces reliance on a gold standard and allows for more accurate evaluations. ELO represents the performance gap between different methods rather than absolute performance values. By comparing ELO scores with real papers, we can approximate the quality of generated ideas by each method.
>
> **References**
>
> [1] Si, Chenglei, Diyi Yang, and Tatsunori Hashimoto. "Can llms generate novel research ideas? a large-scale human study with 100+ nlp researchers." arXiv preprint arXiv:2409.04109 (2024).

---

> ### Author Response · Authors · 2024-11-18
> **Response for Lack of Limitation section and Comparison with SciAgents**
>
> >**Lack of Limitation section 1** : The authors should discuss more on the limitations of the proposed method. Clarify what are achieved and what are not. What else do we need to improve idea generation to achieve the human-level.
>
> **Response**: First of all, we would like to remind the reviewer that limitation section is not mandatory.
> As we have shown via the experimental results, our generated ideas excel in innovation but have a lower level of feasibility compared with the human baseline, as they have not been experimentally validated [1]. Enabling agents to autonomously conduct experiments and refine ideas based on results would complete the research cycle, crucial for automated scientific innovation. This is a future direction for our research, aiming to ensure both innovation and practical feasibility in scientific research automation.
>
> >**Lack of Limitation section 2**: The CoI’s performance heavily depends on the capability and quality of the underlying LLM, which may vary significantly based on the LLM's training and data scope. The authors may test the same task on local LLMs (i.e. weaker LLMs).
>
> **Response**: Generating research ideas is a challenging task and it requires extensive logical analysis and inference. Our goal is not to prove that LLMs can generate ideas, as shown in Yang Diyi’s paper [1], but to explore the upper limit of idea generation by large models. Therefore, we chose the best available model, GPT-4o, and compared it to baselines also based on GPT-4o. Nonetheless, we plan to conduct such experiments with other LLMs in the future.
>
> >**Lack of Limitation section 3**: Although marketed as budget-friendly ($0.50/idea), detailed cost analysis, including potential computing resources and any scaling issues, would provide more transparency into real-world applicability.
>
> **Response**: The specific cost is approximately \$0.1 for generating each idea in a creative chain, an additional \$0.1 for selecting the best idea, and \$0.1 for generating and refining the experiment, totaling \$0.5. You can control the number of chains to manage the cost.
>
> >**Lack of Limitation section 4**: While the focus on AI topics for evaluation demonstrates success, the applicability of the CoI approach in other diverse domains remains unexplored and would benefit from broader validation.
>
> **Response**: The CoI theory is theoretically applicable in any field. We chose the artificial intelligence sector primarily because it is our area of expertise. By selecting AI papers as test samples, we can achieve a better evaluation quality on the generated ideas. With respect to the the core team, there are 2 master's degree holders and 8 PhD holders, with research areas spanning computer vision, embodied intelligence, and natural language processing. Each member has published papers in top international conferences.
> Meanwhile, we have initiated a collaboration with a few psychology professors on using CoI to derive interesting psychological research topics. Before pursuing any of the research ideas, the professors will ensure that all potential ethical concerns are addressed.
>
> >**Comparison with SciAgents**: The concurrent work SciAgents (https://arxiv.org/abs/2409.05556) incorporates graph-based reasoning for idea generation. The authors should compare the proposed method with SciAgents.
>
> **Response**: Thank you for your reminder. Since SciAgents was work conducted concurrently, we did not include it in our paper. After reproducing the results, the specifics are as follows:
> | | Novelty | Significance | Clarity | Feasibility | Effectiveness | Average | Rank |
> |------------------|---------|--------------|---------|-------------|---------------|---------|------|
> | Real Paper| 1073| 1091 | 1161| 1184| 1141| 1130| 1|
> | CoI Agent (ours) | 1156| 1169 | 1092| 1049| 1181| 1129| 2|
> | AI-Researcher| 1133| 1088 | 1106| 1044| 1103| 1095| 3|
> | GPT-Researcher| 993 | 1020 | 1015| 1045| 1021| 1019| 4|
> | ResearchAgent| 1007| 1049 | 1032| 957 | 1038| 1017| 5|
> | RAG| 888| 911| 985 | 1040| 937 | 952 | 6|
> | SciAgent| 891 | 857| 822| 840| 769 | 836 | 7|
> | AI-Scientist | 858 | 815| 788 | 841 | 811| 822| 8|
>
> SciAgent：https://arxiv.org/abs/2409.05556；
> AI-Researcher：https://arxiv.org/abs/2409.04109
>
> **References**
>
> [1] Si, Chenglei, Diyi Yang, and Tatsunori Hashimoto. "Can llms generate novel research ideas? a large-scale human study with 100+ nlp researchers." arXiv preprint arXiv:2409.04109 (2024).

---

> ### Author Response · Authors · 2024-11-18
> **Response for questions**
>
> >**Question 1**:How does the CoI agent handle domains or topics with limited literature or less-established research progression? Does it hallucinate?
>
> **Response**:  Our approach distinguishes itself from other RAG methods by not requiring a large number of relevant papers to achieve significant improvements, as shown in section 4.5. With just 3 papers, noticeable enhancements are realized, and with 5, the results are excellent. In new fields with limited references, such as generating research ideas with LLMs, our method still proves effective, as illustrated in Table 3. However, if no relevant papers are found, or only one is available, our method defaults to the original RAG approach. While hallucination is an inherent issue in LLMs, our strategy significantly mitigates this through RAG by utilizing citation-based retrieval and vector similarity re-ranking, which markedly improves the relevance of retrieved documents and the quality of the Idea Chain. CoI reduces hallucinations by guiding LLMs with an Idea Chain derived from actual research trends, based on sequential existing papers, not generated content. Despite this, the potential for hallucinations in future idea generation remains, prompting us to plan further experiments for more comprehensive evaluation.
>
> >**Question 2**: The current work only considers linear chains. Have you considered more complex chains (multi-path or networked)? Could the CoI framework be adapted to incorporate more dynamic literature, such as rapidly evolving fields where foundational works are continuously supplanted?
>
> **Response**: Our framework employs a dynamic retrieval method from Semantic Scholar, allowing us to track the latest research developments. We intend to explore more complex chain structures in future work, but currently, this paper is to validate the effectiveness of a single chain.
>
> >**Question 3**: What metrics or thresholds determine the construction of a CoI and its length, and how do these parameters affect the quality and novelty of generated ideas?
>
> **Response**: We have carefully evaluated the balance between generation effectiveness and cost. The current chain length and chain numbers we are using were determined through experimentation. Further increasing these variances does not significantly enhance performance but does result in higher costs. These details can be found in our section 4.5，4.6.
>
> >**Question 4**: Is there a systematic way for the CoI agent to identify and mitigate biases in the literature selection that could potentially skew the ideation process?
>
> **Reponse**: We are not entirely sure what "bias" refers to in this context. However, we assume you might be asking how we ensure the quality of the chain. Refer to the response for **Limited analysis and results 2**.
>
> >**Question 5**: Where are the entire results including all the generated ideas?
>
> **Response**: We have show a case in Table 3. The complete results and intermediate processes are not convenient to display due to length issues. For more detail, you can run our code for the full results.
>
> >**Question 6**: Have you considered using graph-based RAG or Knowledge-Graph based approaches?
>
> **Response**: We have considered further research in this area. Previously, we tried a basic graph-based approach (using GoI instead of CoI), but it underperformed compared to the chain method, possibly due to the increased reasoning complexity for the LLM. We plan to refine this by gradually introducing GoI, emulating GoT with graph-like searching and chain-like reasoning.
>
>
> >**Question 7**: Some good services like connected papers can handle a network-based literature survey. Have you considered using their APIs?
>
> **Response**: We have attempted similar services  such as connected-paper(https://github.com/ConnectedPapers/connectedpapers-py), but the quota provided was insufficient for large-scale experiments.
>
> >**Question 8**: Have you optimised the prompts? Have you considered optimising multi-agent systems using ADAS (https://arxiv.org/abs/2408.08435) or AFlow (https://arxiv.org/abs/2410.10762)?
>
> **Response**: Thanks for your reminder. We may consider these methods to improve the quality of our prompts, thereby enhancing the quality of our idea generation.
>
>
> In summary, we appreciate the reviewer's careful review of our paper and the hints that could be helpful for us to explore in the future. We are looking forward to more in-depth discussions with the reviewer in the next few days. Thank you.
>
> **References**
>
> [1] Si, Chenglei, Diyi Yang, and Tatsunori Hashimoto. "Can llms generate novel research ideas? a large-scale human study with 100+ nlp researchers." arXiv preprint arXiv:2409.04109 (2024).

---

> ### Author Response · Authors · 2024-11-19
> **More experimental results on CoT and RAG baseline**
>
> Dear Reviewer wMda:
>
> >**Overall Weakness**: While the chain structure is useful, it may be seen as a slight modification or straightforward adaptation of known LLM prompting strategies, such as chain-of-thought or retrieval-based prompting.
>
> **Response**:Regarding your above question, in addition to our initial responses, we are pleased to provide  additional experiments to distinguish  our method from CoT and retrieval-based prompting (refer to our RAG baseline).
> We compare CoI against three baselines: RAG, CoT, and  CoT + RAG. RAG is the baseline that we already used in our paper, which additionally includes titles and abstracts of 10 papers related to the topic in the input prompt. CoT explicitly asks an LLM to think step-by-step to draft an idea solely based on a given topic.  The CoT + RAG approach combines both strategies, allowing the LLM to think step-by-step while providing it with the titles and abstracts of relevant papers. Similar to Section 4.2, we present the winning score of these three baselines against the original CoI Agent, where each method competed against the CoI Agent in 50 matches, earning 2 points for a win, 1 point for a tie, and 0 points for a loss, with a maximum possible score of 100 points. The following results indicate that  CoT,  RAG, and even CoT + RAG, are less effective than CoI in generating high-quality ideas. This experimental finding further highlights the distinctiveness of our CoI approach in comparison to CoT and retrieval-based prompting.
>
> |  | Novelty | Significance | Clarity | Feasibility |  Effectiveness |
> |-----------|-------|--------|----------------|------------| -------|
> | **CoI(Ours)** | 50.0| 50.0  | 50.0 | 50.0 | 50.0 |
> | **CoT** | 4.0| 1.0  | 14.0 | 25.0 | 0.0 |
> | **RAG** | 8.0|7.0  | 23.0 | 44.0 | 7.0 |
> | **CoT + RAG**| 20.0 | 21.0 | 40.0  | 48.0 | 18.0|

---

> ### Author Response · Authors · 2024-11-22
> **Looking forward to your comments on our response**
>
> Dear reviewer wMda,
>
> We have carefully address the concerns in your review and would greatly appreciate any further comments and suggestions you may have at your earliest convenience.
>
> Thank you.

---

> > ### Author Response · Authors · 2024-11-25
> > **[URGENT] Looking forward to your reply on our response**
> >
> > Dear reviewer wMda,
> >
> > I hope this message finds you well. The discussion period is ending soon, I am writing to emphasize the importance of your review for our submission. Your score is among the lowest of reviewers, and we believe this discrepancy may indicate a misunderstanding or oversight.
> >
> > We have addressed all the concerns in our detailed rebuttal and would appreciate your prompt attention to it. A thorough reassessment is crucial to ensure a fair evaluation.
> >
> > Your expertise is highly valued, and we trust that a reconsidered review will reflect the true merit of our work.
> >
> > Thank you for your immediate attention to this matter.
> >
> > Best regards, Authors

---

> ### Author Response · Authors · 2024-11-29
> **Looking forward to your reply**
>
> Dear wMda,
>
> It has been over ten days since we submitted our rebuttal, and we haven't received your response. We sincerely hope you will join our  discussion, as we highly value the opportunity to engage with you during the rebuttal process. Your feedback is crucial to the improvement of our work, and we have made dedicated efforts to address your concerns. We kindly ask you to take some time to participate in our discussion and share your invaluable insights.
>
> Thank you for considering our request, and we look forward to your response.
>
> Best regards,
>
> Authors

---

### Author Response · Authors · 2024-11-20
**Looking forward to your comments on our response**

Dear reviewers wMda and AfFY,

We have carefully addressed the points in your review and are eager to hear any further comments and suggestions you may have.

Thank you.

---

### Comment · Area_Chair_tHKB · 2024-11-24
**Public discussion phase ending soon**

Dear reviewers,

Thank you for your diligent work on the reviews. Currently the paper has split scores: 5 by reviewers wMda and E21D, 6 by 6L2R and 8 by AfFY.

Reviewer E21D: did the authors' rebuttals and other reviews affect your score? Please respond before the 26 November to let the authors have time to respond if you need any clarification. Thank you!

Your AC

---

### Author Response · Authors · 2024-12-04
**Summary of Reviewer Feedback and Rebuttal Response**

Dear AC，

Thank you for dedicating your time to chair our submission and for your assistance in the discussions!

Overall, we received four reviews with ratings of 8 (AfFY), 6 (6L2R ),5 (wMda) and 5 (E21D). reviewers acknowledged that the proposed CoI agent is novel, significant and effective (wMda, AfFY, 6L2R), the experiments are solid (E21D, wMda), the results are good (wMda, AfFY, E21D) and is budget-friendly (AfFY, 6L2R).

In the rebuttal period, we address the key reviewer concerns as follows:

* *The reliablity of Idea Arena evaluation (wMda, AfFY, E21D)*: We calrify that the 70% human-model agreement is high, considering human-human agreement is only 71.9%. Reviewers AfFY and E21D accepted this explanation, while Reviewer wMda did not reply to us.

* *The difference between CoI and CoT (wMda, E21D)*: we show the difference from both methodological and experimental points. But the two reviewers did not provide further feedback.

* *The task focus (AfFY, 6L2R)*: We clarify why we focus on idea generation rather than the entire paper generation. Both 2 reviewers agreed and raised scores.

* *Ethics (E21D)*: Our work is to improve the idea generation capability for researchers' usage. Safety and hallucination are not the main focus. This is also agreed by reviewer AfFY. We conduct additional safety tests and demonstrate our agent can prevent unsafe idea.

We sincerely appreciate the valuable insights provided by the reviewers and thank you again for your diligent efforts in managing the review process and attention to our work.

Best regards,

Authors

---

### Meta-Review · Area_Chair_tHKB · 2024-12-23

**Metareview:**

The paper introduces a Chain-of-Ideas (CoI), a method to help large language models generate novel research ideas. CoI organizes relevant research papers in a chain structure to simulate the progression of ideas in a field. This helps the models to generate new ideas by identifying current advancements; the authors also propose an evaluation framework, Idea Arena, to assess idea generation from multiple perspectives. The CoI agent outperforms some existing methods and is also cost-effective.

Strengths:
CoI outperforms existing methods in generating research ideas.
The method is cost-effective.
A novel evaluation framework called Idea Arena is proposed.

Weaknesses:
Evaluation metrics, particularly using LLMs as judges, need better justification.
Limited analysis of the generated ideas.
The CoI method might be seen as a simple adaptation of existing prompting techniques.
Potential limitations and biases of the method are not adequately addressed.
Comparison with other relevant work is lacking (addressed with additional experiments in the rebuttal).

Based on the review scores for that paper (5, 5, 5, 8), the paper is just below the acceptance bar.

**Additional Comments On Reviewer Discussion:**

The whole review process was filled with drama, with a record number of 45 posts. The authors have written to complain about wMda for not responding (fair point, although the authors posted ), about E21D for using an LLM to write the review (it does look so) and about 6L2R for violating academic integrity (I do not think so).

---

### Decision · Program_Chairs · 2025-01-22

Reject